# WHEN REASONING MEETS COMPRESSION: UNDERSTANDING THE EFFECTS OF LLMS COMPRESSION ON LARGE REASONING MODELS

**Nan Zhang**♣     **Eugene Kwek**♣     **Yusen Zhang**♣     **Ngoc-Hieu Nguyen**♣
**Prasenjit Mitra**◇     **Rui Zhang**♣
♣The Pennsylvania State University     ◇Carnegie Mellon University Africa
{njz5124,rmz5227}@psu.edu

## ABSTRACT

Compression methods, including quantization, distillation, and pruning, improve the computational efficiency of large reasoning models (LRMs). However, existing studies either fail to sufficiently compare all three compression methods on LRMs or lack in-depth interpretation analysis. In this paper, we investigate how the reasoning capabilities of LRMs are compromised during compression, through performance benchmarking and mechanistic interpretation. To uncover the effects of compression on reasoning performance, we benchmark quantized, distilled, and pruned DeepSeek-R1 models on four reasoning datasets (AIME 2024, FOLIO, Temporal Sequences, and MuSiQue). To precisely locate compression effects on model weights, we adapt difference of means and attribution patching techniques, focusing on the activation of every linear component in compressed LRMs, to interpret fine-grained causal relationships between weights and various reasoning capabilities. This fine-grained interpretation addresses a fundamental question of compression: which weights are the most important for reasoning? Overall, we find dynamically quantized 2.51-bit R1 reaches close-to-R1 performance. With empirical verification, we present three main findings that generalize across both R1 and non-R1 LRMs: (1) Weight count has a greater impact on LRMs' knowledge memorization than reasoning, highlighting the risks of pruning and distillation; (2) The MLP up projection in the final layer of distilled LRMs is one of the most important components, offering a new perspective on locating critical weights - a fundamental problem in model compression; and (3) Current quantization methods overly compress the final-layer modules and MLP gate projections, so protecting just 2% of all weights that are excessively compressed can raise average accuracy by 6.57%, greatly surpassing the state-of-the-art.

## 1 INTRODUCTION

Large reasoning models (LRMs) such as DeepSeek-R1 (Guo et al., 2025) excel at complex reasoning tasks. However, due to their large sizes, deploying them can be costly and even infeasible for individuals, which hinders AI democratization. Compression methods including quantization, distillation, and pruning reduce computational resources (*e.g.*, GPU memory and disk space). Representative quantization techniques include dynamic quantization by Unsloth (Daniel Han & team, 2023), activation-aware quantization AWQ (Lin et al., 2024), and post-training quantization GPTQ (Frantar et al., 2022). Current distillation involves black-box (Li et al., 2024a) or white-box (Gu et al., 2024) settings. Representative pruning techniques include unstructured (Zhang et al., 2024a; Frantar & Alistarh, 2023) and structured pruning (Xia et al., 2024; Ma et al., 2023).

However, existing works do not sufficiently study the performance of compression method on LRMs (Liu et al., 2025a; Srivastava et al., 2025; Feng et al., 2025). Although current quantization and pruning methods claim to preserve the performance of general-purpose LLMs, benchmarking both of them on LRMs with more reasoning-intensive datasets helps compare their collapse point. Regarding distillation, recent works either fail to comprehensively evaluate their student models on diverse reasoning benchmarks of varying difficulty or neglect to consider distillation effect on

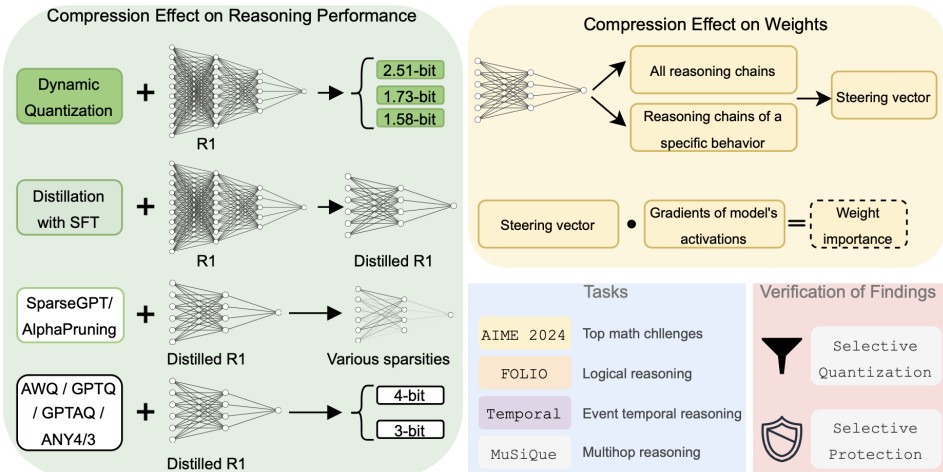

Figure 1: An overview of our pipeline. On the left, we benchmark compressed R1 variants on various reasoning tasks. On the right, by computing weight importance towards a specific reasoning behavior (a dot product of the steering vector and gradients with respect to an LRM's activations), we study the compression effects on individual weight matrices. We empirically verify our findings on weight importance by selectively quantizing or protecting a module to test its importance.

knowledge and reasoning (Huang et al., 2024; Agarwal et al., 2024). Another research gap is the lack of interpretability of compression effects on LRMs. It is necessary to interpret how compression methods affect LRMs, as such analysis can reveal existing bottlenecks and provide guidance for future compression research.

Therefore, due to the lack of compression works on LRMs, we study this fundamental research question: **How are the reasoning capabilities of LRMs compressed during compression?** We answer it from two perspectives: performance benchmarking and mechanistic interpretation. In the main text, we first benchmark compressed DeepSeek-R1 on various reasoning tasks to investigate how model compression affects performance. We test dynamic quantization (Daniel Han & team, 2023), distillation with supervised fine-tuning (SFT) (Guo et al., 2025), SparseGPT (Frantar & Alistarh, 2023), AlphaPruning (Lu et al., 2024), AWQ (Lin et al., 2024), GPTQ (Frantar et al., 2022), GPTAQ (Li et al., 2025), and ANY4/3 (Elhoushi & Johnson, 2025) on R1 (or distilled R1). Then, we apply mechanistic interpretability to quantify weight contribution towards four core reasoning capabilities of LRMs: backtracking, uncertainty estimation, example testing, and adding knowledge. By focusing on the activation of every linear component in compressed LRMs, we adapt difference of means (Arditi et al., 2024) to extract steering vectors and attribution patching (Syed et al., 2023) to compute weight importance. Unlike previous analysis (Venhoff et al., 2025) that only measures layer-wise weight contribution, our weight importance scores offer more fine-grained interpretation of weight contribution, addressing the fundamental compression question of locating important weights. By comparing weight importance between compressed and original LRMs, we quantify the effects of distillation, quantization, and pruning on model weights[1]. Our analysis framework is in Figure 1.

With empirical verification, our key findings are summarized below for better understanding and improving LRMs compression (these findings also generalize to non-R1 families):

- **Weight count** has a greater impact on LRMs' knowledge memorization than their reasoning capabilities, highlighting the compression effects of pruning and distillation. Thus, both distillation and pruning are discouraged when tasks require LRMs' parametric knowledge.
- **The `mlp.up_proj` in the final layer** of R1 distilled models emerges as one of the most important model components, addressing a core concern in pruning and quantization literature: identifying critical weights. Quantizing only this matrix to 3-bit reduces the average accuracy by 16.3%, which validates its high importance.

---

[1]Our interpretation code is at `https://github.com/psunlpgroup/Compression-Effects`.

- **Final-layer modules, along with the `mlp.gate_proj`** of R1 distilled Llama and Qwen, are overly compressed by popular quantization methods, highlighting the need for greater attention to preserving their weight precision. A successful protection of only final-layer MLP modules could raise average accuracy by 6.57%, with gains of up to 23.17% over the state-of-the-art quantization. This key finding also applies to current pruning methods.

## 2 PROBLEM FORMULATION

### 2.1 BACKGROUND

As discussed in Section 1, compression on LRMs (not LLMs) is relatively underexplored. We conduct a thorough literature review in Appendix B.

**Bottlenecks on evaluation.** Few quantization or pruning methodologies have sufficiently demonstrated effectiveness on LRMs. Current works evaluate quantization and pruning performance primarily using perplexity and simple end tasks, such as the EleutherAI evaluation harness (Gao et al., 2024) and commonsense reasoning. However, quantized or pruned LRMs should be assessed on more complex reasoning tasks with varying difficulty levels. For distillation, although recent works tend to test on more challenging reasoning tasks (compared to other compression literature) such as GSM8K (Cobbe et al., 2021), it is unclear how the compression of LRMs affects models' parametric knowledge and reasoning capability. Some of them do not comprehensively select diverse reasoning benchmarks (Agarwal et al., 2024). Our benchmarking aims to address these bottlenecks.

**Bottlenecks on in-depth analysis.** The lack of interpretability of compression effects on LRMs is a key bottleneck of in-depth analysis for compressed LRMs. Being able to interpret the difference between original and compressed LRMs offers a new way to analyze the effects of compression. As a result, better compression approaches can be developed. A recent work (Venhoff et al., 2025) interprets several R1 distilled LRMs, but their focus is not on understanding compression effects.

**Recent efforts.** Recent benchmarking (Liu et al., 2025a) and survey (Feng et al., 2025; Srivastava et al., 2025) papers have begun to evaluate compressed LRMs on more complex reasoning datasets, but they all lack in-depth interpretation of compression effects and do not comprehensively compare different compression strategies. As for compressed LRMs, Unsloth (Daniel Han & team, 2023) introduces dynamic quantization by dynamically opting not to quantize certain LLM weights. DeepSeek-R1 (Guo et al., 2025) also comes with several distilled models via black-box distillation. Our interpretation analysis aims to demystify the effects of LLMs compression on LRMs, providing a systematic understanding of existing compressed LRMs.

### 2.2 MECHANISTIC INTERPRETATION

For our interpretation analysis, we target four core reasoning behaviors following an existing work (Venhoff et al., 2025): backtracking, uncertainty estimation, example testing, and adding knowledge. We prompt GPT-4o to locate token sequences of each behavior from the output tokens of our LRMs. Our annotation dataset consists of 120 instances drawn from the four benchmark datasets (30 instances from each), so our interpretability analysis spans four different task types and difficulties. Annotation robustness of GPT-4o is demonstrated in Appendix G.

To interpret different compression strategies, we adapt difference of means and attribution patching by computing the activation of every linear module in each layer. This allows us to compute the causal relationship between each weight matrix and our target reasoning behaviors.

**Difference of Means.** To compute the numerical representation in activation space of each reasoning behavior, we adapt difference of means method (Venhoff et al., 2025; Arditi et al., 2024) to extract the steering vector $\mathbf{u}_{m\ell}^c$ for each linear module $m$ at layer $\ell$ for behavior $c$:

$$\mathbf{u}_{m\ell}^c = \frac{1}{|\mathcal{D}_+|} \sum_{s_i^c \in \mathcal{D}_+} \overline{\mathbf{a}}_{m\ell}^c(s_i^c) - \frac{1}{|\mathcal{D}_-|} \sum_{s_j \in \mathcal{D}_-} \overline{\mathbf{a}}_{m\ell}^c(s_j), \quad \text{with} \quad \overline{\mathbf{a}}_{m\ell}^c(s_i^c) = \frac{1}{|s_i^c|} \sum_{t \in s_i^c} \mathbf{a}_{m\ell}(t)$$

where $s_i^c$ denotes the token sequence corresponding to a specific reasoning behavior $c$ along with its five preceding tokens as output by an LRM, $s_j$ is the token sequence of the entire LRM output (prompt and output tokens), $\mathcal{D}_+$ is the set of output instances containing at least one token sequence

labeled with $c$, $\mathcal{D}_-$ is the set of all output instances, $\mathbf{a}_{m\ell}(t)$ is the activation of module $m$ at layer $\ell$ at token $t$, $\overline{\mathbf{a}}_{m\ell}^c(s_i^c)$ is the average of $\mathbf{a}_{m\ell}(t)$ across all tokens in $s_i^c$, and similarly, $\overline{\mathbf{a}}_{m\ell}(s_j)$ is the average of $\mathbf{a}_{m\ell}(t)$ across all tokens in $s_j$. We then normalize $\mathbf{u}_{m\ell}^c$ to $\tilde{\mathbf{u}}_{m\ell}^c$: $\tilde{\mathbf{u}}_{m\ell}^c = \mathbf{u}_{m\ell}^c \cdot \frac{\|\overline{\mathbf{a}}_{m\ell}^{\mathrm{all}}\|_2}{\|\mathbf{u}_{m\ell}^c\|_2}$ where $\overline{\mathbf{a}}_{m\ell}^{\mathrm{all}}$ denotes the mean activation across all tokens in $\mathcal{D}_-$.

**Attribution Patching.** To find the causally relevant LRMs components with respect to each reasoning behavior, we adapt attribution patching (Syed et al., 2023) method to compute the importance score $\mathbf{I}_{m\ell}^c$ of each linear module.

$$\mathbf{I}_{m\ell}^c \approx \frac{1}{|\mathcal{D}_+|} \left| \sum_{s_i^c \in \mathcal{D}_+} \left( \tilde{\mathbf{u}}_{m\ell}^c \right)^\top \frac{\partial}{\partial \mathbf{a}_{m\ell}} \mathcal{L}(s_i^c) \right|$$

where $\mathcal{L}(s_i^c)$ is the cross-entropy loss of $s_i^c$. A higher $\mathbf{I}_{m\ell}^c$ means a stronger causal relationship between $c$ and the linear module $m$ at layer $\ell$, helping us locate the most important weights responsible for reasoning capabilities (a fundamental problem for quantization and pruning works).

## 2.3 DECODING COMPRESSION EFFECTS

To decode compression effects, we compute the relative importance $\mathbf{RI}_{m\ell}^c$ of each weight matrix ($\mathbf{I}_{m\ell}^c$ divided by $\sum_m \sum_\ell \mathbf{I}_{m\ell}^c$) and track how it changes because of compression (**importance shift**). Specifically, we measure the change of $\mathbf{RI}_{m\ell}^c$ from R1 distilled Llama-8B to original `meta-llama/Llama-3.1-8B` to understand distillation effect (Section 4). Likewise, the importance shift from the R1 distilled models to their quantized versions indicates quantization effect (Section 5). For the R1 distilled models, we also compute the $\mathbf{I}_{m\ell}^c$ of their weights to complement our findings on the distillation effect.

We hypothesize that the importance shift should be minimal in the ideal case, as a compressed LRM should remain as close as possible to its original counterpart (the more reasoning-capable model). **When visualizing the importance shift from an LRM to its compressed variant (or from a distilled model to its backbone), we only consider decreases in $\mathbf{RI}_{m\ell}^c$.** By definition, the relative importance of each weight matrix is normalized to sum to one, so any increase in relative importance necessarily compensates for decreases elsewhere. Since it is more informative to track cases where the $\mathbf{RI}_{m\ell}^c$ of a more reasoning-capable model decreases (*e.g.*, when the reasoning capability of a weight matrix is diminished), we set all increases in relative importance to zero. Additional justification of only visualizing the decreases is in Appendix H.

## 2.4 SCOPE

We study three major LLMs compression paradigms, distillation, quantization, and pruning, making our scope comprehensive enough for investigating the effects of diverse compression methods. For distillation, we select four R1 distilled models: `DeepSeek-R1-Distill` Llama-70B, Qwen-32B, Llama-8B, and Qwen-7B. For quantization, we select 2.51-, 1.73-, and 1.58-bit models by Unsloth[2] as the choices of quantized R1 due to their popularity. We also evaluate AWQ Lin et al. (2024), GPTQ (Frantar et al., 2022), GPTAQ (Li et al., 2025), and ANY4/3 (Elhoushi & Johnson, 2025) as reproducible state-of-the-art quantization methods designed for relatively smaller LLMs (*e.g.*, the R1 distilled models). Specifically, we use all four methods to perform 4-bit quantization, and use GPTQ, GPTAQ, and ANY3 for 3-bit quantization as well, since many AWQ implementations do not support 3-bit. For pruning, we run SparseGPT Frantar & Alistarh (2023) and AlphaPruning (Lu et al., 2024) on several distilled models. Inside AlphaPruning, SparseGPT is applied to prune the layers after layerwise compression ratios are computed. We run interpretation analysis on linear modules of all layers within LRMs.

## 2.5 EVALUATION SETUP

We select four reasoning datasets with varying levels of difficulty: AIME 2024 (Mathematical Association of America) for mathematical reasoning, FOLIO (Han et al., 2024) for logical reasoning, Temporal Sequences of BIG-Bench Hard (Suzgun et al., 2022) for temporal reasoning, and

---

[2]`https://huggingface.co/unsloth/DeepSeek-R1-GGUF`

MuSiQue (Trivedi et al., 2022) for multihop reasoning. Since MuSiQue requires knowledge memorization besides multihop reasoning, we follow a closed-book setting (directly prompting LRMs to get final answers) to evaluate both reasoning and knowledge retention capabilities. Additional details of benchmarks, along with Table 5 that shows their statistics, are specified in Appendix C.

Accuracy metric is used for AIME 2024, FOLIO, and Temporal Sequences. We adopt exact match (EM) and F1 for MuSiQue. For each model (except R1 and those dynamically quantized LRMs), we run it three times and report its average scores to mitigate performance variability. Implementation details are in Appendix D.

## 3    COMPRESSION EFFECTS ON REASONING PERFORMANCE

In this section, we examine the effects of compression on reasoning performance. Interestingly, our key findings from the following sections also generalize to non-R1 model families, as elaborated in Appendix J.

### 3.1    OVERALL PERFORMANCE

The overall performance of R1 and its compressed variants are in Table 1. We show the performance of pruned `R1-Distill-Llama-70B` and `R1-Distill-Qwen-32B` under 50% sparsity in Table 1, as it is the default sparsity level of current works (Zhang et al., 2024a; Sun et al., 2023). Additional analysis of test-time compute and model collapse behavior are in Appendices O and P.

**Comparing Compression Strategies.** In Table 1, the 2.51-bit R1 achieves the highest average accuracy overall, since it has the smallest compression ratio. Both R1 distilled Llama-70B and Qwen-32B reach close-to-R1 accuracy scores. On MuSiQue, the 2.51-bit R1 also achieves performance close to original R1. Therefore, 2.51-bit R1 has the best overall performance than other compression strategies. Although R1 may be over-parameterized, a compression method with a smaller ratio can still offer advantages over methods with higher compression ratios. In contrast, pruning only 50% of the weights causes significant degradation, rendering the pruned LRMs unusable. Thus, we choose to interpret the effect of pruning with greater caution and specify the details in Appendix I. As for all distillation-only models, Qwen delivers stronger reasoning performance than Llama (Appendix E).

**Comparing Benchmark Difficulties.** Comparing the scores using R1 distilled Llama-70B on AIME 2024, FOLIO, and Temporal, we see the largest score decrease on AIME 2024. This indicates that AIME 2024 is more challenging than the other two accuracy-based benchmarks. MuSiQue is also difficult in terms of knowledge requirement, because its scores in Table 1 are much lower than RAG (retrieval-augmented generation) setup (Zhang et al., 2025). This suggests that existing LRMs lack sufficient knowledge for knowledge-intensive tasks, making RAG a more suitable approach.

> **Takeaway 3.1 for Overall Performance**
>
> Considering over-parameterization, methods with smaller compression ratios can still offer advantages over those with higher compression ratios. Regardless of whether compression is applied, LRMs lack sufficient knowledge for knowledge-intensive tasks.

### 3.2    COLLAPSE POINT

We investigate whether LRMs degrade as they undergo increasing levels of compression. In Table 1, the performance of dynamically quantized LRMs steadily declines as we move from 2.51 to 1.58-bit, but we do not observe a clear collapse point. All 4-bit AWQ, GPTQ, GPTAQ, and ANY4 reach performance similar to their unquantized counterparts, which shows the effectiveness of existing 4-bit quantization on LRMs. However, 3-bit GPTQ, GPTAQ, and ANY3 display signs of collapse, indicating bottlenecks of current 3-bit quantization. GPTAQ and ANY4 are newer than AWQ and GPTQ, and they achieve similar performance on 4-bit LRMs. Based on average accuracy, it is noteworthy that GPTAQ surpasses GPTQ on 3-bit LRMs for three out of four distilled models. Regarding distillation, R1 distilled Llama-8B and Qwen-7B achieve the lowest accuracy among all distillation-only models. Only R1 distilled Llama-70B yields decent MuSiQue scores.

Table 1: Benchmark performance of R1 and its compressed variants. All four benchmark scores are averaged over three passes, except the rows marked with [†]. Avg denotes the average scores shown in AIME 2024, FOLIO, and Temporal columns. We segment this table based on model families and mark the highest scores within each model family in **bold**.

| Models | | | Accuracy | | | | |
|---|---|---|---|---|---|---|---|
| Model | #Param | Compression | AIME 2024 | FOLIO | Temporal | Avg | MuSiQue (EM, F1) |
| DeepSeek-R1[†] | 671B | - | 73.3 | 76.4 | 99.6 | 83.1 | (**17.0**, **27.51**) |
| DeepSeek-R1[†] | 671B | 2.51-bit | **76.7** | 77.8 | **100.0** | **84.8** | (**17.0**, 24.43) |
| DeepSeek-R1[†] | 671B | 1.73-bit | 66.7 | **78.3** | 99.6 | 81.5 | (15.0, 22.11) |
| DeepSeek-R1[†] | 671B | 1.58-bit | 66.7 | 75.4 | 94.0 | 78.7 | (14.0, 22.34) |
| R1-Distill-Llama | 70B | Distillation | 65.6 | **79.8** | **99.9** | **81.8** | (**13.3**, **21.57**) |
| R1-Distill-Llama | 70B | Distillation & 50% SparseGPT | 23.3 | 71.6 | 97.6 | 64.2 | (6.7, 13.49) |
| R1-Distill-Llama | 70B | Distillation & 50% AlphaPruning | 26.7 | 74.2 | 97.7 | 66.2 | (5.3, 12.39) |
| R1-Distill-Llama | 70B | Distillation & 4-bit AWQ | 63.4 | 78.5 | 99.3 | 80.4 | (10.7, 19.23) |
| R1-Distill-Llama | 70B | Distillation & 4-bit GPTQ | **66.7** | 77.0 | **99.9** | 81.2 | (10.3, 18.17) |
| R1-Distill-Llama | 70B | Distillation & 4-bit GPTAQ | 64.4 | 78.8 | 99.6 | 80.9 | (12.0, **21.57**) |
| R1-Distill-Llama | 70B | Distillation & 3-bit GPTQ | 46.7 | 71.8 | 99.3 | 72.6 | (4.7, 11.92) |
| R1-Distill-Llama | 70B | Distillation & 3-bit GPTAQ | 54.4 | 77.3 | 99.7 | 77.1 | (5.7, 13.21) |
| R1-Distill-Qwen | 32B | Distillation | 64.4 | 82.3 | **99.9** | 82.2 | (2.7, 10.95) |
| R1-Distill-Qwen | 32B | Distillation & 50% SparseGPT | 25.6 | 75.1 | 97.9 | 66.2 | (2.3, 9.01) |
| R1-Distill-Qwen | 32B | Distillation & 4-bit AWQ | 67.8 | 82.3 | 99.1 | **83.1** | (3.3, 10.28) |
| R1-Distill-Qwen | 32B | Distillation & 4-bit GPTQ | **68.9** | 80.6 | 99.6 | 83.0 | (4.0, 11.78) |
| R1-Distill-Qwen | 32B | Distillation & 4-bit GPTAQ | 63.3 | 81.5 | 99.7 | 81.5 | (2.7, 11.88) |
| R1-Distill-Qwen | 32B | Distillation & 4-bit ANY4 | **68.9** | 78.0 | 99.7 | 82.2 | (**5.7**, **12.68**) |
| R1-Distill-Qwen | 32B | Distillation & 3-bit GPTQ | 44.4 | 74.2 | 98.9 | 72.5 | (4.0, 11.55) |
| R1-Distill-Qwen | 32B | Distillation & 3-bit GPTAQ | 45.6 | 77.5 | 99.5 | 74.2 | (2.3, 9.18) |
| R1-Distill-Qwen | 32B | Distillation & 3-bit ANY3 | 53.3 | **82.6** | **99.9** | 78.6 | (3.7, 10.27) |
| R1-Distill-Llama | 8B | Distillation | 42.2 | **71.9** | 81.5 | 65.2 | (0.0, 4.43) |
| R1-Distill-Llama | 8B | Distillation & 30% AlphaPruning | 41.1 | 68.9 | 82.1 | 64.1 | (0.3, 4.51) |
| R1-Distill-Llama | 8B | Distillation & 50% AlphaPruning | 6.7 | 61.7 | 79.6 | 49.3 | (0.0, 2.95) |
| R1-Distill-Llama | 8B | Distillation & 4-bit AWQ | **47.8** | 68.0 | 84.0 | **66.6** | (**0.3**, **5.05**) |
| R1-Distill-Llama | 8B | Distillation & 4-bit GPTQ | 42.2 | 66.2 | 65.9 | 58.1 | (**0.3**, 4.68) |
| R1-Distill-Llama | 8B | Distillation & 4-bit GPTAQ | 40.0 | 66.4 | 69.3 | 58.6 | (0.0, 3.73) |
| R1-Distill-Llama | 8B | Distillation & 4-bit ANY4 | 41.1 | 68.5 | **88.7** | 66.1 | (0.0, 3.54) |
| R1-Distill-Llama | 8B | Distillation & 3-bit GPTQ | 11.1 | 65.0 | 67.3 | 47.8 | (0.0, 2.89) |
| R1-Distill-Llama | 8B | Distillation & 3-bit GPTAQ | 7.8 | 65.5 | 57.2 | 43.5 | (0.0, 3.45) |
| R1-Distill-Llama | 8B | Distillation & 3-bit ANY3 | 3.3 | 50.1 | 34.9 | 29.4 | (0.7, 2.35) |
| R1-Distill-Qwen | 7B | Distillation | 46.7 | **78.0** | 75.6 | **66.8** | (0.0, 3.57) |
| R1-Distill-Qwen | 7B | Distillation & 4-bit AWQ | 46.6 | 75.5 | 74.9 | 65.7 | (0.0, 3.14) |
| R1-Distill-Qwen | 7B | Distillation & 4-bit GPTQ | 38.9 | 72.9 | 70.3 | 60.7 | (**1.0**, **4.27**) |
| R1-Distill-Qwen | 7B | Distillation & 4-bit GPTAQ | **47.8** | 74.4 | 67.7 | 63.3 | (0.0, 3.96) |
| R1-Distill-Qwen | 7B | Distillation & 4-bit ANY4 | **47.8** | 75.6 | **77.1** | **66.8** | (0.0, 3.05) |
| R1-Distill-Qwen | 7B | Distillation & 3-bit GPTQ | 17.8 | 65.7 | 31.7 | 38.4 | (0.0, 3.12) |
| R1-Distill-Qwen | 7B | Distillation & 3-bit GPTAQ | 24.4 | 64.5 | 48.7 | 45.9 | (0.0, 3.06) |
| R1-Distill-Qwen | 7B | Distillation & 3-bit ANY3 | 32.2 | 69.3 | 30.1 | 43.9 | (0.0, 3.89) |

Table 2 displays performance of our two distilled models under various sparsity levels. Comparing distilled models with their sparsified variants, we find the precise collapse points of our pruned LRMs. Interestingly, their collapse points correlate to the benchmark difficulty. For example, on AIME 2024, `R1-Distill-Llama` collapses between 40% and 50% sparsity, since its performance drops by more than half. However, its collapse points on FOLIO and Temporal are roughly between 60% and 70% sparsity, which occur much later than AIME 2024. The correlation between collapse point and benchmark difficulty can also be seen on the sparsified Qwen.

**Takeaway 3.2 for Collapse Point**

> Collapse point correlates with benchmark difficulty. On hard benchmarks, 3-bit quantization and pruning with 50% sparsity or higher still have substantial room for improvement.

## 3.3 COMPRESSION IMPACT ON KNOWLEDGE AND REASONING

In Table 2, although Qwen demonstrates stronger reasoning capabilities than Llama, it has significantly lower EM and F1 scores on MuSiQue. Because MuSiQue requires knowledge memorization under the closed-book setting, the smaller parameter count of Qwen puts itself at a disadvantaged position. In other words, models' parameter count affects knowledge more than reasoning. In addi-

Table 2: Performance of two distilled models under various sparsity levels of SparseGPT. We report the one-pass scores for all models in this table.

| Models | | | Accuracy | | | | |
|---|---|---|---|---|---|---|---|
| Model | #Param | Sparsity | AIME 2024 | FOLIO | Temporal | Avg | MuSiQue (EM, F1) |
| R1-Distill-Llama | 70B | 0% | 63.3 | 78.8 | 100.0 | 80.7 | (13.0, **21.80**) |
| R1-Distill-Llama | 70B | 10% | 60.0 | 81.3 | 99.6 | 80.3 | (12.0, 21.69) |
| R1-Distill-Llama | 70B | 30% | 63.3 | 79.3 | 99.6 | 80.7 | (**14.0**, 21.40) |
| R1-Distill-Llama | 70B | 40% | 56.7 | 73.9 | 98.8 | 76.8 | (6.0, 13.79) |
| R1-Distill-Llama | 70B | 50% | 26.7 | 70.9 | 97.2 | 64.9 | (6.0, 12.75) |
| R1-Distill-Llama | 70B | 60% | 0.0 | 65.0 | 95.6 | 53.5 | (0.0, 6.42) |
| R1-Distill-Llama | 70B | 70% | 0.0 | 49.8 | 15.6 | 21.8 | (0.0, 2.23) |
| R1-Distill-Llama | 70B | 80% | 0.0 | 11.8 | 12.4 | 8.1 | (0.0, 0.94) |
| R1-Distill-Qwen | 32B | 0% | 66.7 | 82.3 | 100.0 | 83.0 | (1.0, 9.38) |
| R1-Distill-Qwen | 32B | 10% | 70.0 | 81.3 | 100.0 | 83.8 | (**5.0**, **13.19**) |
| R1-Distill-Qwen | 32B | 30% | 56.7 | 81.3 | 100.0 | 79.3 | (1.0, 10.47) |
| R1-Distill-Qwen | 32B | 40% | 53.3 | 78.3 | 100.0 | 77.2 | (2.0, 10.16) |
| R1-Distill-Qwen | 32B | 50% | 30.0 | 75.4 | 96.0 | 67.1 | (3.0, 9.29) |
| R1-Distill-Qwen | 32B | 60% | 0.0 | 65.0 | 87.2 | 50.7 | (0.0, 4.13) |
| R1-Distill-Qwen | 32B | 70% | 0.0 | 32.5 | 19.6 | 17.4 | (0.0, 1.72) |
| R1-Distill-Qwen | 32B | 80% | 0.0 | 8.7 | 2.0 | 3.6 | (0.0, 1.29) |

tion, we notice that pruned `R1-Distill-Llama-70B` collapses between 30% and 40% sparsity on MuSiQue, which is even earlier than on AIME 2024. This shows that pruning hurts LRMs' knowledge memorization more than quantization. When a compression method aggressively removes the weights of an LRM, it is expected that the model's knowledge will be more seriously affected. This phenomenon can also be seen on our dynamically quantized models in Table 1. Since quantization preserves parameter count and our analysis above shows that many quantized models still retain competitive reasoning capability, quantization is recommended on knowledge-intensive tasks. Additional experiments on knowledge retrieval are specified in Appendix L.

> **Takeaway 3.3 for Compression Impact on Knowledge and Reasoning**
>
> Pruning and distillation compress knowledge retention more than reasoning capabilities.

# 4 DISTILLATION EFFECT ON WEIGHTS

To study the effect of distillation on weights, we compute $\mathbf{I}^c_{m\ell}$ of two distilled R1 models and further measure the change of $\mathbf{RI}^c_{m\ell}$ as discussed in Section 2.3.

## 4.1 LOCATING IMPORTANT WEIGHTS

The upper part of Figure 2 presents the weight importance of R1 distilled Llama-8B in four heatmaps, each corresponding to a reasoning behavior. We observe that the final layer houses several most important linear modules across all four behaviors, with the highest value located at `up_proj`. Therefore, the `up_proj` in the final layer (`32_up`) stands out as the most important component.

Interestingly, this finding generalizes to R1 distilled Qwen-7B, as we also observe this `up_proj` outlier in the final layer of Qwen in Figure 4. Notably, our finding complements a recent analysis (Shao & Wu, 2025), which claims the most important module for reasoning is `o_proj`. Since identifying important weights is a core research problem of compression methodologies, our finding is valuable for future works. Additional diagnosis based on AWQ also reinforces our finding on the last-layer up projection (Appendix M).

> **Takeaway 4.1 for Locating Important Weights**
>
> Distillation makes `up_proj` in the final layer as the most important module for reasoning behaviors, as observed in both R1 distilled Llama and Qwen models.

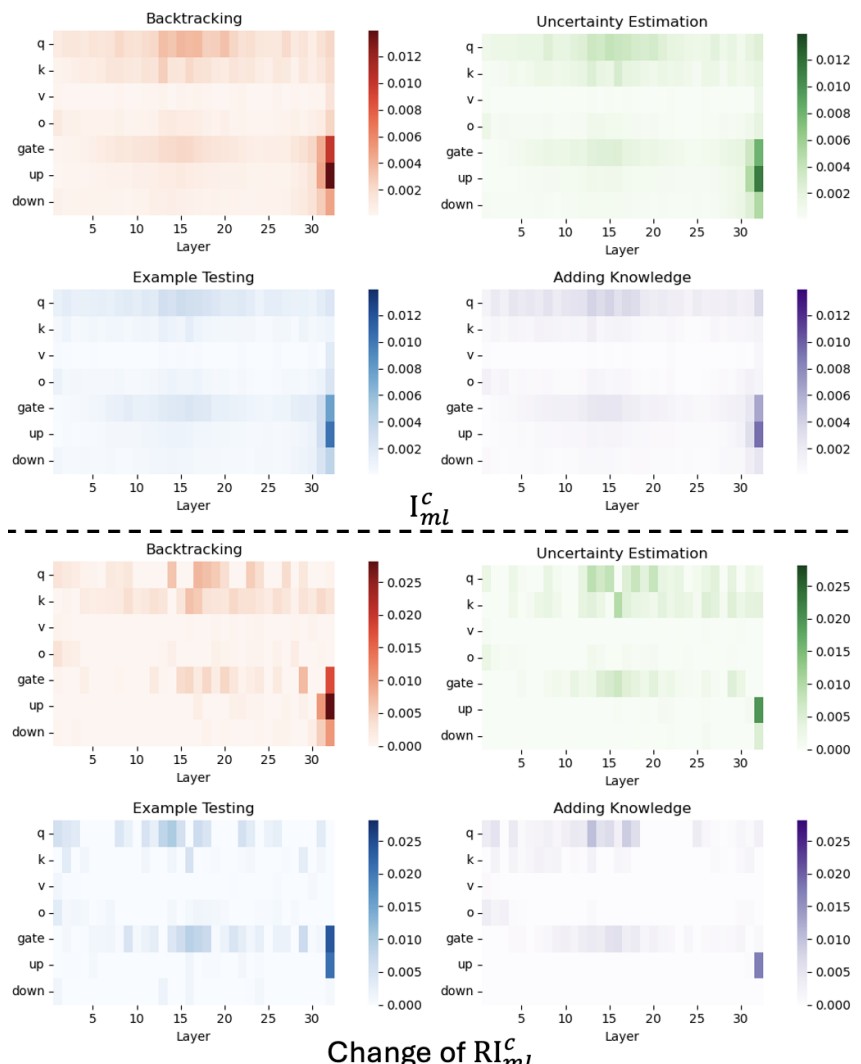

Figure 2: $\mathbf{I}^c_{m\ell}$ of DeepSeek-R1-Distill-Llama-8B (**upper**) and change of $\mathbf{RI}^c_{m\ell}$ from DeepSeek-R1-Distill-Llama-8B to Llama-3.1-8B (**lower**). Each heatmap displays scores of importance (or importance shift) of every module at each layer, providing a fine-grained analysis of weight contributions to the corresponding reasoning capability. For the lower part, increases in $\mathbf{RI}^c_{m\ell}$ are set to 0, as they only offset decreases elsewhere as discussed in 2.3. Every cluster of 4 side-by-side heatmaps (including those displayed below) follow the same scaling to show the precise magnitude of each weight module.

## 4.2 VALIDATING IMPORTANCE SCORES

We validate Section 4.1 by applying 3-bit round-to-nearest quantization to either 32_up or a component sharing its column or row in the heatmaps, then measuring the resulting accuracy drop (Table 3). The more important a component is, the greater the accuracy drop when it is quantized. Specifically, we select four additional component candidates: the second- and last-ranked modules among the seven in the final layer, and the second- and last-ranked layers across all 32 layers of the up-projection. We see that 32_up yields the lowest average accuracy, which clearly demonstrates the validity of our findings in Section 4.1. It is quite salient that quantizing only this matrix (merely 0.7% of all weights) reduces the average accuracy by 16.3%. The component rank generally correlates with the accuracy drop, except for 1_up, which incurs the lowest accuracy on AIME 2024.

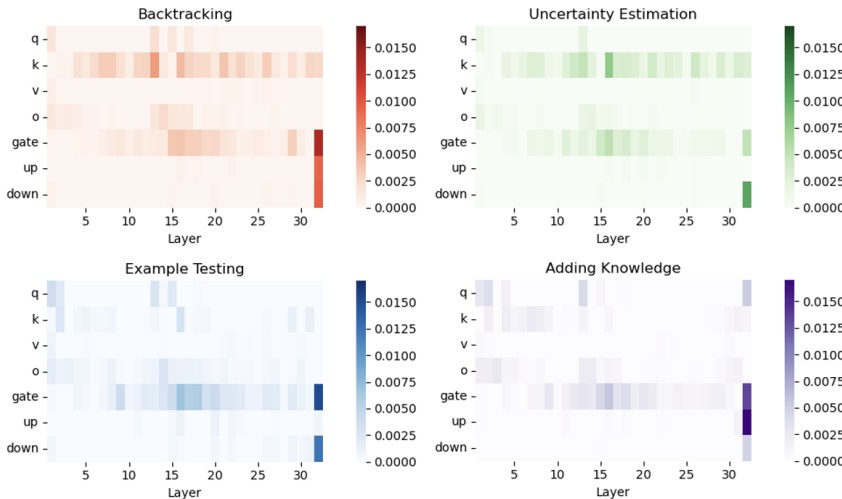

Figure 3: Change of $\mathbf{RI}^c_{m\ell}$ from `DeepSeek-R1-Distill-Llama-8B` to its 4-bit AWQ variant. Justification of only showing the decrease of $\mathbf{RI}^c_{m\ell}$ in the main content is specified in Appendix H.

Table 3: Accuracy after selectively quantizing a single component of R1 distilled Llama-8B (*e.g.*, 1_up means to only quantize the `up_proj` in the first layer) to 3-bit. Ranking of a component is based its $\sum_c \mathbf{I}_{m\ell}$, so "2nd col" refers to second place within its column across all four heatmaps (each column consists of 7 linear modules of a layer). "1st overall" means the global highest ranking. We provide additional validation on AIME 2024 scores in Appendix N.

| Quantized Component | Rank | AIME 2024 | FOLIO | Temporal | Avg |
|---|---|---|---|---|---|
| 32_up | 1st overall | 20.0 | 63.1 | 63.6 | 48.9 |
| 32_gate | 2nd col | 33.3 | 62.1 | 67.2 | 54.2 |
| 32_v | last col | 43.3 | 68.0 | 79.6 | 63.6 |
| 31_up | 2nd row | 33.3 | 70.0 | 64.4 | 55.9 |
| 1_up | last row | 6.7 | 64.5 | 80.4 | 50.5 |

## 4.3 IMPORTANCE SHIFT VIA DISTILLATION

Since `R1-Distill-Llama-8B` is fine-tuned based on `Llama-3.1-8B`, we compute the change of $\mathbf{RI}^c_{m\ell}$ to visualize distillation effect in the lower part of Figure 2 (for Llama) and 5 (for Qwen). Both parts of Figure 2 exhibit similar patterns (*e.g.*, most outliers are in the final layer), indicating that the important weights of the distilled model are primarily the result of distillation with SFT, while the original Llama's weight values play little role in shaping its reasoning capabilities (we elaborate on $\mathbf{I}^c_{m\ell}$ of `Llama-3.1-8B` in Appendix K). Thus, distillation effect is quite powerful in transforming a non-reasoning LLM into an LRM. For Qwen, Figures 4 and 5 also show similar patterns, so the utility of distillation generalizes to Qwen as well. In conclusion, distillation effect explains why the final-layer up projection is one of the most important LRMs components for reasoning.

**Takeaway 4.3 for Importance Shift via Distillation**

Important weights of the R1 distilled models are mainly the result of the distillation effect.

## 5 QUANTIZATION EFFECT ON WEIGHTS

For quantization effect on weights, we analyze the decrease of importance shift during quantization. Pruning effect based on AlphaPruning appears very similar to quantization effect and is specified in Appendix I.

Table 4: Performance of 3-bit AWQ and selectively protecting the MLP modules in the final layer.

| Model | Compression | Full-Precision Anywhere? | AIME 2024 | FOLIO | Temporal | Avg | MuSiQue |
|---|---|---|---|---|---|---|---|
| R1-Distill-Llama-8B | 3-bit AWQ | - | 10.0 | 59.6 | 68.4 | 46.0 | (0.0, 3.50) |
| R1-Distill-Llama-8B | 3-bit AWQ | Final-layer MLP | **16.7** | **67.0** | **74.0** | **52.57** | **(1.0, 3.62)** |

## 5.1 LOCATING QUANTIZATION EFFECT

We show heatmaps to visualize the importance shift from R1 distilled Llama-8B to its 4-bit AWQ quantized variant in Figure 3. Across all four heatmaps, we observe a reduction in the significance of the gate projections in the middle layers (*e.g.*, layer 9 to 23), suggesting that AWQ may overly compress these modules. Moreover, most linear modules in the final layer are compressed to the greatest extent, which shows the drawback of AWQ. Since 32_up is the most important module as discussed in Section 4.2 and its importance shift is little for uncertainty estimation and example testing capabilities, AWQ successfully preserves its significance on these two behaviors. However, for backtracking and adding knowledge capabilities, AWQ is not effective at maintaining its importance.

In Figure 6, we visualize the importance shift from R1 distilled Qwen-7B to its 4-bit AWQ quantized version. We also see a shift in the importance of the gate projections on Qwen, but this shift mainly occurs in the early layers (*e.g.*, layer 1 to 10). On Qwen, AWQ does not preserve the importance of 32_up across all four reasoning capabilities, and it also overly compresses 32_k on two capabilities.

As another popular method, we interpret the effect of 4-bit GPTQ in Figure 7. On R1 distilled Llama-8B, we observe similar quantization effect as AWQ, since GPTQ also overly compresses final-layer modules and the gate projections in the middle layers. Their commonality demonstrates the generality of the bottlenecks we identified in existing quantization methods.

**Takeaway 5 for Quantization Effect on Weights**

State-of-the-art quantization methods fail to preserve the importance of the MLP gate projections and the final layer, which is a key bottleneck of performance improvement.

## 5.2 VALIDATING QUANTIZATION EFFECT

To validate our findings about the bottleneck of current quantization, we design a simple protection mechanism using a mixed precision fashion. We run two versions of 3-bit AWQ in Table 4. In the first version, we run AWQ with their default calibration data. Since we know AWQ overly compresses the MLP modules in the final layer, we then choose to protect them by changing their quantized weights to their original values in 16-bit. If they are truly important yet not well protected by AWQ, our protection mechanism should offer a significant improvement. Based on the discussion in Sections 3.1 and 3.2, we perform 3-bit quantization, since it will be the focus of future works.

We see our selective protection boosts 3-bit AWQ on all benchmarks, with an average accuracy improvement of 6.57%. This is particularly significant given that only about 2% of all weights remain in 16-bit. This mixed precision model outperforms all 3-bit quantization baselines in Table 1 by at least 4.77% in average accuracy, with gains of up to 23.17%. Therefore, our findings are demonstrated with an indication of substantial room for further improvement. Note that our protection provides relatively marginal increase on MuSiQue, as the weight count stays the same (Section 3.3).

## 6 CONCLUSION AND FUTURE DIRECTIONS

We study the effects of LLMs compression on LRMs and present key findings for further improving LRMs compression. These findings generalize across both R1 and non-R1 models. Future compression works are encouraged to consider the protection of MLP up projection in the final layer. The excessive compression of current quantization and pruning methods on MLP gate projections and final-layer modules highlights the need for better preserving these weight modules.

ACKNOWLEDGMENTS

This work was supported by NSF IIS-2338418. We thank Yanchi Liu for his valuable feedback on the project.

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

## A  USE OF LLMS

To improve the overall clarity of our writing, we used ChatGPT-4o and ChatGPT-5 via OpenAI's web interface to polish a small fraction of sentences. LLMs were not used in any steps of the research ideation process. To ensure correctness and precision, we carefully reviewed and adapted all LLM-generated content before incorporating it into our writing.

## B  RELATED WORK

Our literature review is conducted over existing compression methodologies (quantization, distillation, and pruning), LRMs, and recent module-wise compression.

### B.1  QUANTIZATION

Quantization reduces the number of bits used to represent LLM weights, thereby lowering their precision (Srivastava et al., 2025). Recent survey (Zhu et al., 2024) categorizes quantization methodologies into quantization-aware training (QAT) and post-training quantization (PTQ). QAT requires retraining of model weights to recover performance loss during quantization while PTQ does not require retraining. Recent QAT includes LLM-QAT (Liu et al., 2024a) that adopts distillation to

train a quantized LLM, BitDistiller (Du et al., 2024) that develops a self-distillation approach for the full-precision model to act as the teacher of its low-bit counterpart, BitNet (Wang et al., 2023) that proposes a 1-bit Transformer architecture for training LLMs from scratch, and OneBit (Xu et al., 2024) that quantizes LLM weight matrices to 1-bit from a knowledge transfer perspective.

PTQ is more popular in terms of the number of recent publications, because there is no retraining involved. For example, GPTQ (Frantar et al., 2022) and GPTAQ (Li et al., 2025) are one-shot weight quantization methods that use approximate second-order information, while AWQ (Lin et al., 2024) leverages activation distribution for finding the salient weight channels to skip. Other PTQ methods include weight-activation quantization (Shao et al., 2024; Yao et al., 2022; Liu et al., 2023) and KV cache quantization (Hooper et al., 2024; Liu et al., 2024b).

## B.2    DISTILLATION

Distillation involves two settings: black-box and white-box settings. For black-box setting, the teacher model is typically a closed-source LLM and only the outputs of teacher are available for the student model. For white-box setting, both weights and output distribution of the teacher model are available. Existing black-box distillation (Huang et al., 2024; Li et al., 2024b; Ho et al., 2023; Huang et al., 2022; Li et al., 2024a) prompts the teacher model to generate a training dataset for the student to learn. Specifically, researchers have started to distill OpenAI's O1 model (Huang et al., 2024), which marks the beginning of LRMs compression. White-box distillation allows the student model to learn from teacher's knowledge representation. Works has been done to align the output distribution (Agarwal et al., 2024; Gu et al., 2024) or the hidden representation (Liang et al., 2023) between teacher and student models.

## B.3    PRUNING

There are unstructured and structured pruning. For unstructured pruning, individual weights are targeted, which leads to irregular sparsity structure. In contrast, structured pruning involves removing entire network components such as channels or layers (Zhang et al., 2024a). Unstructured pruning usually has better compression performance than structured pruning, while it is easier to achieve inference speedup via structured methods (Zhu et al., 2024). Recent unstructured pruning includes one-shot pruning (Frantar & Alistarh, 2023; Sun et al., 2023), global pruning that makes pruning decisions based on all layers (Bai et al., 2024), and domain-specific pruning (Zhang et al., 2024a). Structured pruning includes gradient-based (Xia et al., 2024; Ma et al., 2023) and non-gradient-based (Ashkboos et al., 2024) methods.

## B.4    LRMS

Trained with reinforcement learning, LRMs extends LLMs with advanced reasoning mechanisms (Besta et al., 2025). Popular closed-source LRMs are OpenAI's o1-mini, o1 (OpenAI et al., 2024), and o3-mini. Open-source LRMs include DeepSeek-R1 and QwQ-32B-Preview (Team, 2024). Since quantization, white-box distillation, and pruning methods require access to model weights, they are not suitable for closed-source LRMs. Only black-box distillation will work on closed-source models.

## B.5    MODULE-WISE COMPRESSION

Recent module-wise compression methods (Lu et al., 2024; Yin et al., 2025) are designed to address imbalanced quality among different model components (*e.g.*, layers and weight matrices). This research direction could potentially address the problem of over-compressing the important weights, which is also one of our motivations of locating important weights. As a way of enriching module-specific compression, we speculate that using non-uniform compression ratios across different modules (*e.g.*, unstructured pruning and mix-precision quantization) might pose challenges for inference speedup. A potential solution of being more hardware-efficient is to enforce uniform ratios with protection mechanisms on important weights. For example, AWQ protects salient weights identified via activation signals by scaling them up, while still using a uniform compression ratio across modules.

Table 5: Dataset statistics of selected reasoning benchmarks.

|  | Size | Answer Type | Metric | Knowledge Required? |
|---|---|---|---|---|
| AIME 2024 | 30 | Integer | Accuracy | False |
| FOLIO | 203 | True/False/Uncertain | Accuracy | False |
| Temporal | 250 | (A)/(B)/(C)/(D) | Accuracy | False |
| MuSiQue | 100 | A few words | (EM, F1) | True |

As the cornerstone of module-wise compression, it is quite valuable to look for novel importance signals for locating salient weights (Liu et al., 2025b). For example, our paper decodes the effects of compression on LRMs and validates the usefulness and generalizability of fine-grained mechanistic interpretation for LRMs compression. A successful compression method might need to incorporate multiple signals or merge them, as some may benefit from a change in the compression goal (*e.g.*, changing the domain or the compression ratio).

## C  ADDITIONAL DETAILS OF REASONING BENCHMARKS

Table 5 shows the statistics of our selected benchmarks. AIME 2024[3] (parts I and II) represents top match challenges, and its answers are integers. FOLIO[4] requires logical deductions to determine whether the provided conclusion is true, false, or uncertain based on premise. In Temporal Sequences[5], models are asked to use a provided timeline to determine what time a person might be free to perform another activity. Since each of its questions comes with four options, we expect our models to output the index (the letter) of the selected option. Since MuSiQue involves question answering and its answers are in a few words, we adopt exact match (EM) and F1. We randomly sample 100 questions out of 1000 from MuSiQue for our benchmarking analysis.

## D  IMPLEMENTATION DETAILS

We run the dynamically quantized models on `llama.cpp`[6] based on their requirement. We run all other distilled, pruned, and quantized models on vLLM (Kwon et al., 2023) for its fast inference. In order to comprehensively analyze performance change after compression, we also evaluate R1 on our reasoning benchmarks by using DeepSeek API. Aligning with DeepSeek-R1 report (Guo et al., 2025), we keep the same parameters for all models during inference: maximum generation length is set to 32768, temperature is set to 0.6, and top-p value is set to 0.95.

Within the 2.51-bit LRM, the embedding matrices, output/head, and normalization/router layers are kept at higher bit widths, while almost everything else is quantized aggressively. Specifically, the embedding matrix is in 4-bit and the final language model head is in 6-bit, while all MoE (Mixture of Experts) routing matrices and layer-norm weights are left in full precision (32-bit). The majority (around 88%) of weights – namely the bulk of the MoE weights – are quantized down to 2.51-bit.

We do not find precise details of the calibration dataset used for 2.51-bit quantization. However, we see that Unsloth plans to use more than 1.5 million tokens for future quantized models, so we can make a reasonable prediction that the dynamic quantization uses more calibration data than other quantization methods we benchmark in our paper. We adopt the same calibration data for all other quantization. The dynamically quantized models work on MoE architecture such as R1, while other quantization methods (*e.g.*, AWQ and GPTAQ) are designed for non-MoE with less weight count. Therefore, due to different compression targets, we are not able to compare dynamic quantization and other methods under identical calibration. In Table 1, we aim to report various methods as comprehensive as possible in order to study the effects of compression on reasoning performance.

---

[3]https://huggingface.co/datasets/Maxwell-Jia/AIME_2024
[4]https://huggingface.co/datasets/yale-nlp/FOLIO
[5]https://github.com/suzgunmirac/BIG-Bench-Hard/blob/main/bbh/temporal_sequences.json
[6]https://github.com/ggml-org/llama.cpp

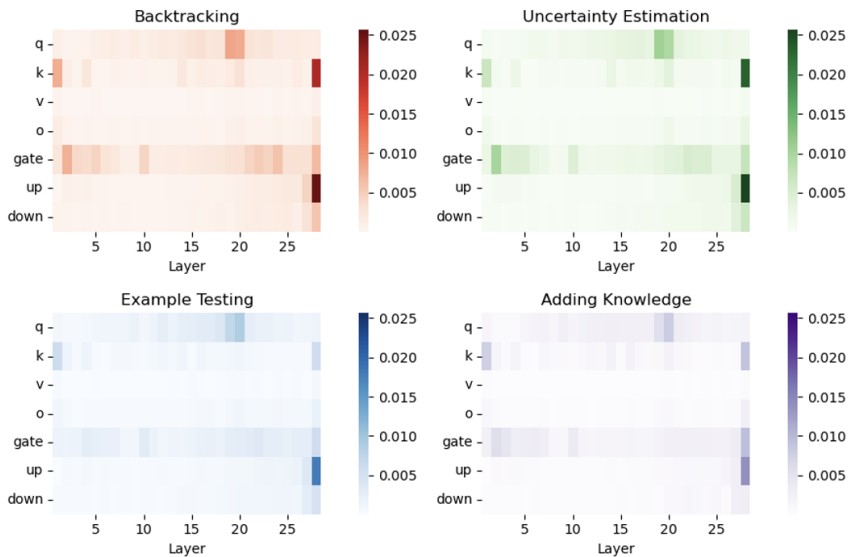

Figure 4: $\mathbf{I}_{m\ell}^c$ of `DeepSeek-R1-Distill-Qwen-7B`.

We use AutoAWQ[7] as the AWQ implementation for inference due to its speed advantage (vLLM support), while the original AWQ code[8] is used to generate the pseudo-quantized R1 distilled Llama-8B for our analysis in Section 5.

We focus on analyzing the effect of compression methods on performance and thus do not consider inference speedup. The reason is that these methods run on different inference platforms, so it is hard to control the consistency of inference optimization across various platforms.

## E    COMPARING DISTILLED MODELS

On accuracy-based benchmarks of Table 1, we see that R1 distilled Qwen-32B delivers an average 0.4% improvement over Llama-70 and R1 distilled Qwen-7B delivers an average 1.6% improvement over Llama-8B. Although these two Qwen models have less weights, Qwen delivers stronger reasoning performance than Llama. This phenomenon aligns with DeepSeek report (Guo et al., 2025). However, `R1-Distill-Qwen-32B` scores significantly lower than `R1-Distill-Llama-70B` on MuSiQue, highlighting its worse ability of memorization.

## F    ADDITIONAL VISUALIZATION OF WEIGHT IMPORTANCE AND IMPORTANCE SHIFT

Figure 4 shows the weight importance of `DeepSeek-R1-Distill-Qwen-7B` across four heatmaps, each corresponding to a specific target reasoning behavior. Figure 5 displays the change of $\mathbf{RI}_{m\ell}^c$ from `DeepSeek-R1-Distill-Llama-8B` to `Qwen2.5-Math-7B`. To decode the quantization effect on Qwen, Figure 6 shows the change of $\mathbf{RI}_{m\ell}^c$ from `DeepSeek-R1-Distill-Qwen-7B` to its 4-bit AWQ variant. Similarly, Figure 7 shows the change of $\mathbf{RI}_{m\ell}^c$ from R1 distilled Llama-8B to its 4-bit GPTQ quantized variant.

## G    ANNOTATION ROBUSTNESS

It is important to find a validated approach to automatically perform annotation, as a large number of $s_i^c$ would help the computation of importance scores. We did not change the behavior definitions and

---

[7]`https://github.com/casper-hansen/AutoAWQ`
[8]`https://github.com/mit-han-lab/llm-awq`

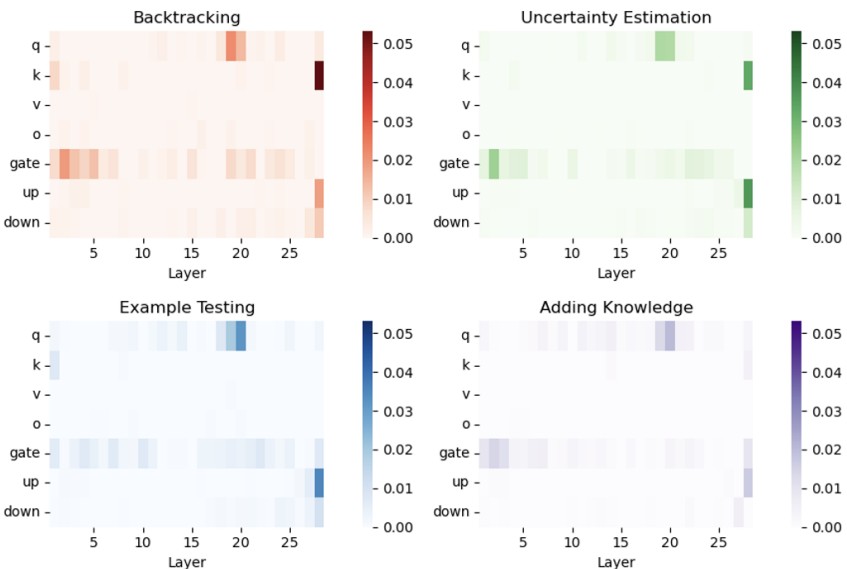

Figure 5: Change of $\mathbf{RI}_{m\ell}^c$ from `DeepSeek-R1-Distill-Llama-8B` to `Qwen2.5-Math-7B` (the backbone model).

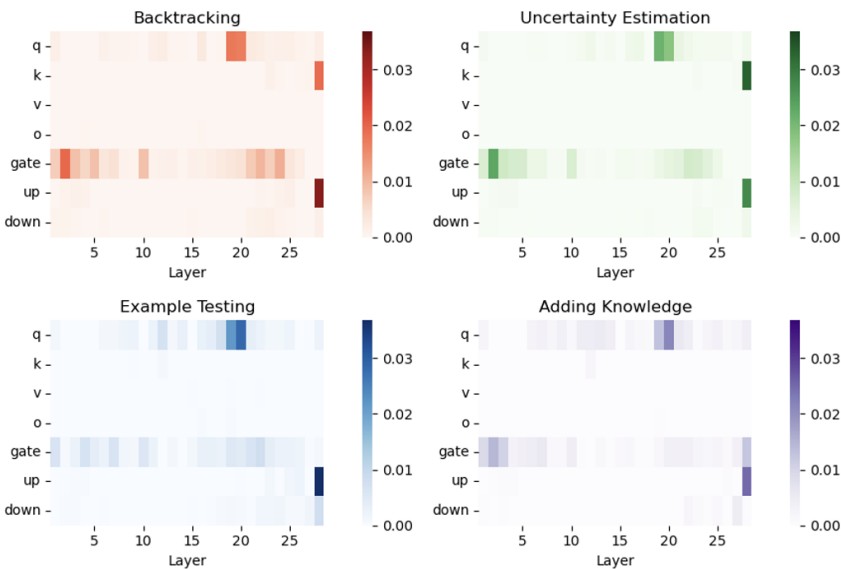

Figure 6: Change of $\mathbf{RI}_{m\ell}^c$ from `DeepSeek-R1-Distill-Qwen-7B` to its 4-bit AWQ variant.

Table 6: Statistics of ratings under 2 temperature values. "CI" represents 95% confidence interval.

| Behavior | Avg (temp=1) | Avg (temp=0.1) | Spearman $\rho$ | CI |
|---|---|---|---|---|
| Backtracking | 0.9 | 0.94 | 0.892 | [0.474, 1.000] |
| Uncertainty Estimation | 0.78 | 0.75 | 0.803 | [0.416, 1.000] |
| Example Testing | 0.90 | 0.87 | 0.862 | [0.645, 1.000] |
| Adding Knowledge | 0.86 | 0.96 | 0.486 | [-0.218, 1.000] |

instruction prompts. The most salient adjustable parameter is the temperature, and we used the de-

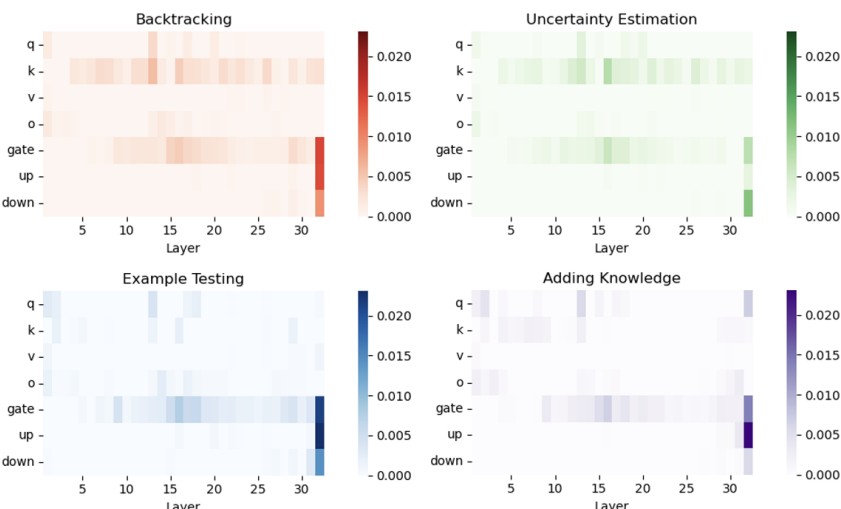

Figure 7: Change of $\mathbf{RI}^c_{m\ell}$ from `DeepSeek-R1-Distill-Llama-8B` to its 4-bit GPTQ quantized variant.

fault temperature value (1) in our annotation pipeline. We study whether the change of temperature would overturn our annotation results.

We choose a low temperature value (0.1) and see whether more deterministic outputs would affect our annotation results significantly. Setting temperature to 0.1, we first use GPT-4o with the exact same behavior definitions and instruction prompts to collect annotation. Then, we perform human validation to judge the output quality and stability when a lower temperature is set. To score a model output instance on a specific behavior, we instruct the evaluator to go through all the behavior tokens that are tagged and compute the percentage of correctly tagged behavior. For example, suppose GPT-4o tags 20 "backtracking" token sequences in an output instance and the evaluator finds 18 of them closely following the definition of "backtracking", the score of this instance is 0.9. There are cases when an instance ends up with no tagged behavior. Since we encourage a conservative annotation style (the value of $s_i^c$ is large enough for computation such as 1000, so it is fine if GPT-4o misses some token sequences of the target behavior), we assign the score of 1 (maximum score) to those instances ending up with 0 tagged behavior. For each behavior, the evaluator samples 10 instances out of 120 (with replacement) and scores them. We present the average score of 10 sampled instances under 2 temperatures and the Spearman $\rho$ with 95% confidence interval in Table 6.

As shown, the average rating scores of both temperatures are high across 4 behaviors, which indicates reliable annotation results done by GPT-4o. Comparing the results of both temperatures side-by-side, we observe that a temperature of 1 tends to produce more nested and unclosed token sequences, which is expected due to higher randomness. This does not affect our interpretation analysis, since we exclude those nested or unclosed sequences.

As for Spearman $\rho$, we see that the two rating scores on all 4 reasoning behaviors exhibit at least a moderate and often strong correlation, which demonstrates stable rank ordering. For "adding knowledge" behavior, the 95% confidence interval is wide, indicating uncertainty due to the small sample size of 10. Given high average rating scores, we conclude using the default temperature value does not significantly affect the correctness of annotation. Therefore, the robustness of our findings is validated.

## H  JUSTIFICATION OF VISUALIZATION CHOICE

As elaborated in Section 2.3, we track changes in relative importance from a more reasoning-capable model to a less reasoning-capable model (rather than the other way around). Then, we mention to only consider decreases to show our interest in tracking the cases when the reasoning capability of a weight matrix is diminished. In other words, when an LRM is compressed via suboptimal

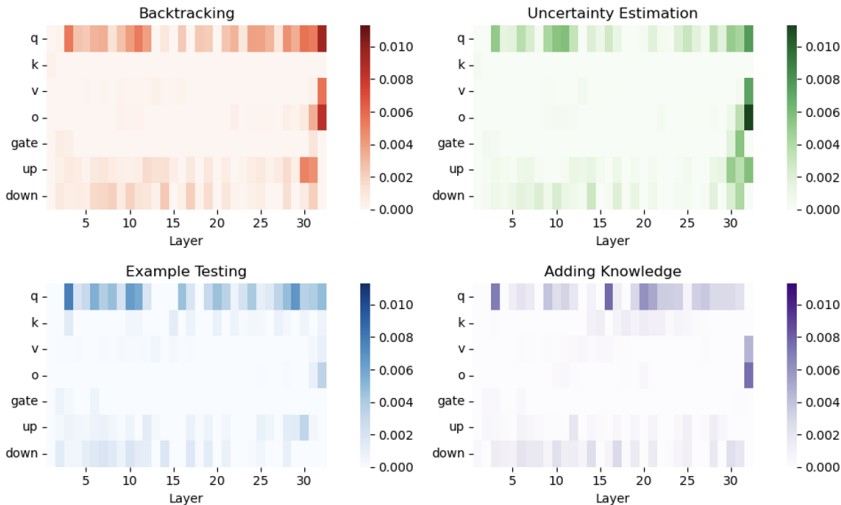

Figure 8: Change of $\mathbf{RI}^c_{m\ell}$ (**increases only**) from `DeepSeek-R1-Distill-Llama-8B` to its 4-bit AWQ variant.

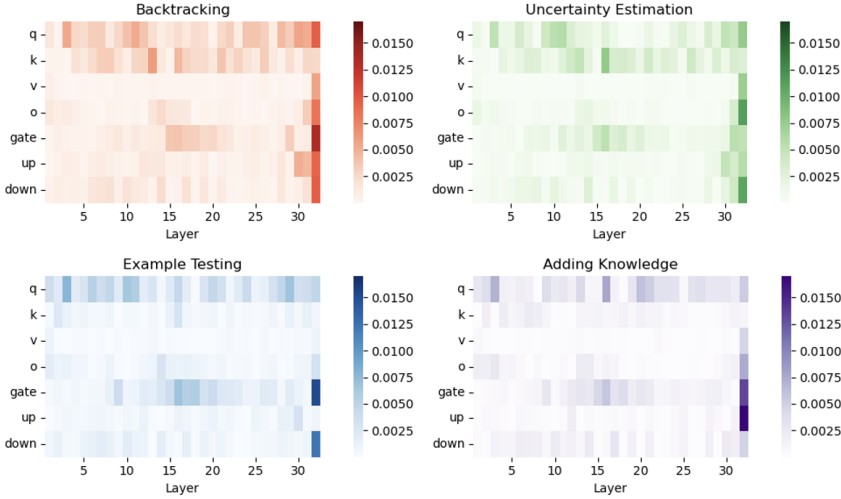

Figure 9: Change of $\mathbf{RI}^c_{m\ell}$ (**net change**) from `DeepSeek-R1-Distill-Llama-8B` to its 4-bit AWQ variant.

pruning or quantization, salient weight components are more likely to show decreased rather than increased relative importance, because suboptimal compression tends to preserve or harm, but not improve, the reasoning capabilities of important components. For compression methods that fully preserve or increase the relative importance of some salient components, tracking these increases adds little value, as it essentially confirms that the methods are performing well. It is more valuable to visualize the effects of compression with an emphasis on the weakness of current compression methods, which can inspire future improvement such as our third key finding.

On the other hand, in order to show all aspects of importance shift, we draw the heatmaps of only considering increases (Figure 8) and those that reflect the net change (Figure 9) from R1 distilled Llama-8B to its 4-bit AWQ variant. We see that the net-change heatmaps appear quite similar to those that only consider decreases (Figure 3). Specifically, the net-change heatmaps primarily highlight only a few components from the increase-only heatmaps based on Figure 3, and our third key finding still holds (researchers are recommended to prioritize the preservation of final-layer-module importance in order to improve current quantization methods). The similarity between Figure 3 and

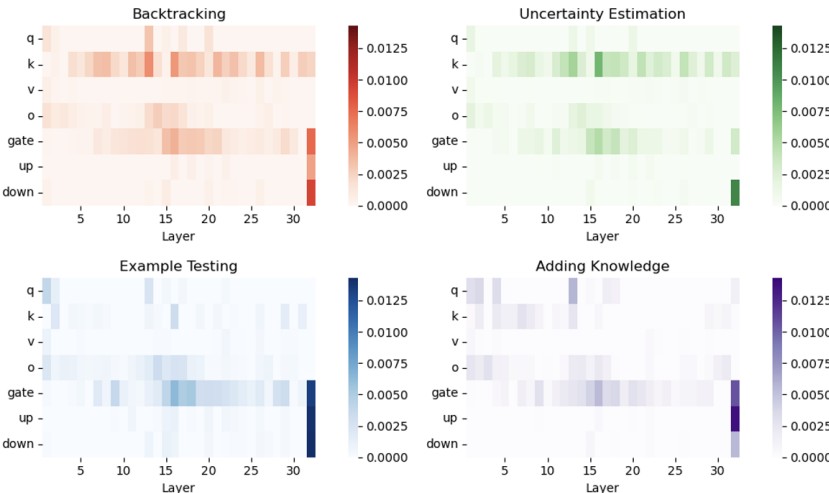

Figure 10: Change of $\mathbf{RI}_{m\ell}^c$ from `DeepSeek-R1-Distill-Llama-8B` to its AlphaPruning variant (at 30% sparsity).

net-change heatmaps showcases a significant shortcoming of current quantization, as most of the net change comes from diminishing capabilities of certain salient modules in Figure 3.

## I   PRUNING EFFECT ON WEIGHTS

As mentioned in Section 3.1, we require a pruned LRM that remains usable for interpretability analysis, as annotating and interpreting outputs that consist largely of gibberish would be meaningless. Therefore, in order to provide more insights on pruning, we turn to AlphaPruning (compared to SparseGPT on the R1 distilled Llama-70B, AlphaPruning achieves higher scores on most benchmarks in Table 1). On R1 distilled Llama-8B, we iteratively reduce the 50% sparsity level and ultimately settle on 30%, where much stronger reasoning performance is achieved. Although 30% sparsity (1.4x reduction in size) offers much smaller compression ratio than 4-bit quantization (4x reduction in size), its "avg" score of 64.1 is significantly higher than 4-bit GPTQ and GPTAQ in Table 1. We then select AlphaPruning at 30% sparsity on the R1-distilled Llama-8B as the pruned LRM for our interpretability analysis. We show the corresponding heatmaps in Figure 10.

We observe that Figure 10 appears very similar to 4-bit AWQ on R1 distilled Llama-8B (Figure 3), 4-bit GPTQ on R1 distilled Llama-8B (Figure 7), and 4-bit AWQ on R1 distilled Qwen-7B (Figure 6). Therefore, our third key finding can be nicely generalized to AlphaPruning as well: AlphaPruning overly compresses the final-layer modules and MLP gate projections. This additional experiment shows the effects of pruning and the generalizability of our third key finding on pruning methods.

## J   GENERALIZABILITY OF KEY FINDINGS ON NON-R1 FAMILIES

We are excited to share that our key findings generalize well to `Pinkstack/DistilGPT-OSS-qwen3-4B` (a distilled model from both `GPT-OSS-20B` and `GPT-OSS-120B` via SFT). In order to systematically investigate weight importance and quantization effect, we interpret this distilled model and its 4-bit GPTQ quantized variant. We first show their one-pass benchmarking scores in Table 7. It is clear to see that MuSiQue scores are not affected by quantization, further implying that preserving weight count with lower precisions is a reasonable strategy for LRMs' knowledge memorization (first key finding).

Then, we look at our second key finding. After running our fine-grained interpretation analysis on `DistilGPT-OSS-qwen3-4B`, we present its heatmaps in Figure 11. We see that 36_up (up_projection in the final layer) is either the most important or the second most important component across four heatmaps, which demonstrates the generalizability of our second key finding.

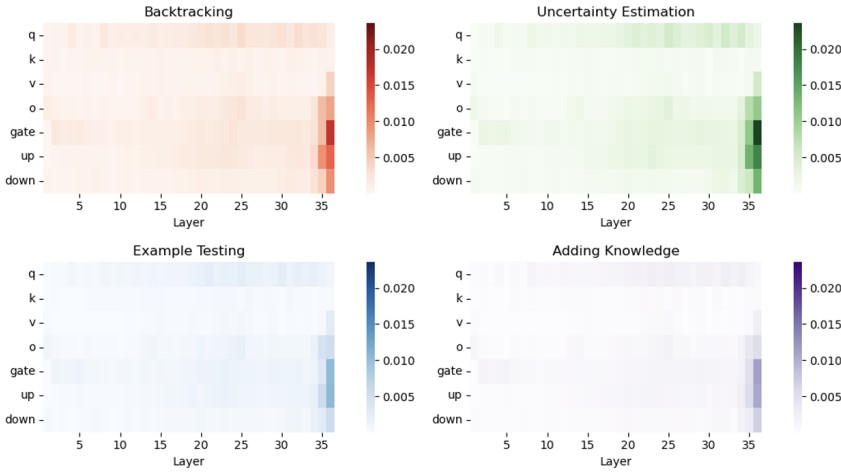

Figure 11: $\mathbf{I}^c_{m\ell}$ of `DistilGPT-OSS-qwen3-4B`.

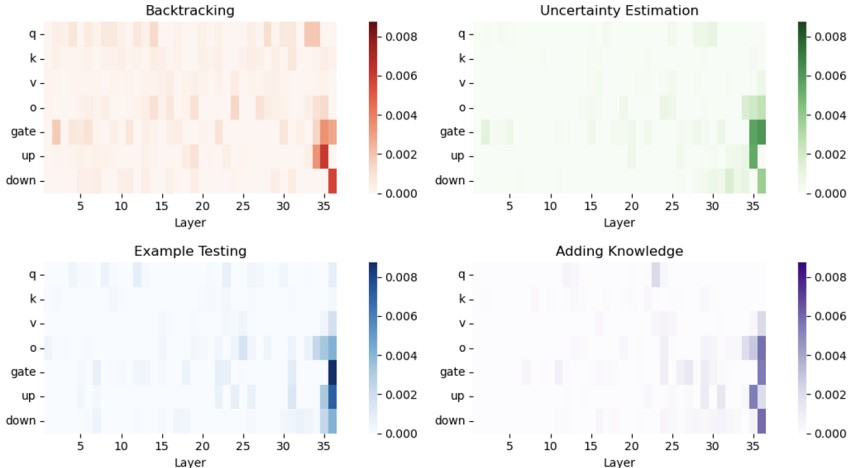

Figure 12: Change of $\mathbf{RI}^c_{m\ell}$ from `DistilGPT-OSS-qwen3-4B` to its 4-bit GPTQ variant.

Table 7: Performance of `DistilGPT-OSS-qwen3-4B` and its 4-bit GPTQ variant.

| Model | AIME 2024 | FOLIO | Temporal | Avg | MuSiQue |
|---|---|---|---|---|---|
| DistilGPT-OSS-qwen3-4B | 36.7 | 82.3 | 94.4 | 71.1 | (4.0, 6.73) |
| 4-bit GPTQ quantized DistilGPT-OSS-qwen3-4B | 26.7 | 76.4 | 86.4 | 63.2 | (4.0, 6.92) |

Moreover, whenever 36_up is ranked second on a heatmap, 36_gate is always ranked first. As both up_projection and gate_projection in the final layer are important to R1 distilled models in Figure 2 (upper part) and 4, we see a high similarity of weight importance between R1 distilled models and this GPT-OSS distilled model, which indicates the possibility of finding common important components across different LRMs. On the other hand, it is worth noting that different LRMs might end up with slightly different heatmaps. For example, last-layer k_projection is highlighted in Figure 4 but appears to be less important for the GPT-OSS distilled model. This shows that our interpretation can produce customized analysis of weight importance. Our second key finding aims to locate the import weight component that is shared by different LRMs (*e.g.*, different model sizes and classes).

Finally, for our third key finding, we demonstrate its generalizability through the 4-bit GPTQ quantized `DistilGPT-OSS-qwen3-4B`. The heatmaps for visualizing decrease of relative importance

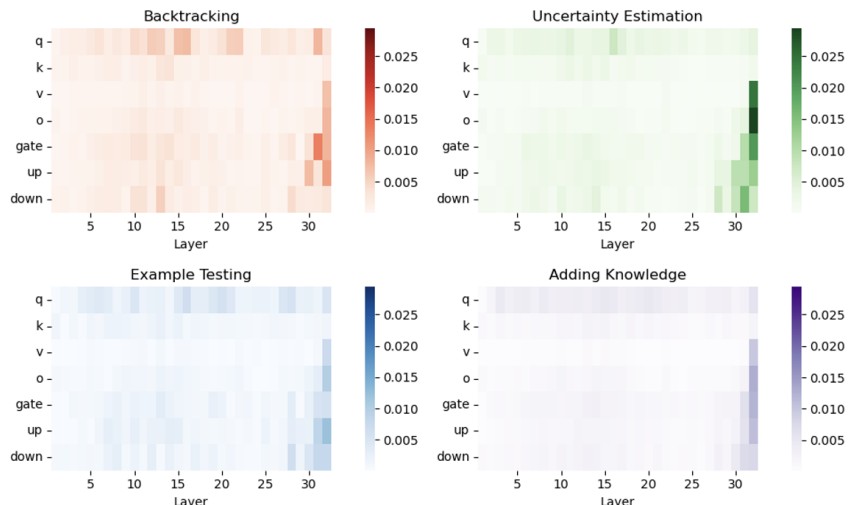

Figure 13: $\mathbf{I}^c_{m\ell}$ of `Llama-3.1-8B`.

Table 8: Performance of naive RAG and closed-book settings on MuSiQue.

| Setting | LRM | MuSiQue |
|---|---|---|
| Closed-book | R1-Distill-Llama-8B | (0.0, 4.43) |
| RAG | R1-Distill-Llama-8B | (2.0, 14.52) |
| Closed-book | R1-Distill-Qwen-7B | (0.0, 3.57) |
| RAG | R1-Distill-Qwen-7B | (2.0, 13.54) |

are shown in Figure 12. Most salient decreases (outliers) come from the last-layer components across all four heatmaps, along with a few salient decreases in the second last layer. Out of a small number of outliers in a heatmap, gate_projection also has at least 2 outliers on the backtracking and uncertainty estimation capabilities. Therefore, even when we switch to a GPT-OSS distilled model, GPTQ still overly compresses the final-layer modules and MLP gate projections.

## K    IMPORTANCE SCORES OF BASE MODEL

Complementing the lower part (change of relative importance from R1 distilled Llama-8B to its base model before distillation) of Figure 2, we present the importance scores of the base model (`Llama-3.1-8B`) in Figure 13. As we can see, the heatmaps of the base model look very different from those of the R1-distilled model (upper part of Figure 2). There are many differences. For example, compared to uncertainty estimation behavior, other reasoning behaviors do not present a clear outlier of importance, which indicates more evenly distributed causal relationships across weight matrices on backtracking, example testing, and adding knowledge. As for uncertainty estimation, the last-layer o_projection has the highest importance, whereas the up_projection in the final layer is not even ranked among the top three components of that layer. Therefore, we could reach a completely different conclusion from key finding 2 if we were to interpret the base model rather than the LRMs. The commonality between the two parts of Figure 2, together with the difference between the base model and the lower part of Figure 2, demonstrates Takeaway 4.3 ("important weights of the R1 distilled models are mainly the result of the distillation effect").

## L    ADDITIONAL RAG PIPELINE

Section 3.3 concentrates on the significantly lower MuSiQue scores as we move from R1 to different compressed variants. We hypothesize that parametric knowledge is more sensitive to parameter count, because pruned R1 distilled Llama-70B in Table 2 collapses on MuSiQue even earlier than

Table 9: Performance of selectively quantizing 32_up or 1_up on AIME 2024.

| Selective Quantization | 32_up | 1_up |
|---|---|---|
| 4-bit (R1-Distill-Llama-8B) | 20.0 | 43.3 |
| 5-bit (R1-Distill-Llama-8B) | 26.7 | 30.0 |
| 3-bit (R1-Distill-Qwen-7B) | 16.7 | 46.7 |

AIME 2024. Other notable phenomena include the slightly stronger reasoning performance of R1 distilled Qwen-32B than Llama-70B, but with collapsed MuSiQue scores at even 0% sparsity.

In order to further validate our key finding 1, we implement a RAG pipeline across two R1 distilled models with different sizes. Since all MuSiQue scores in Table 1 follow the closed-book setting, we provide results from naive RAG to decouple reasoning from knowledge memorization. Specifically, we use `BAAI/bge-base-en-v1.5` as the embedding model and retrieve the top 10 chunks whose embeddings are closest to each question in MuSiQue. We run this RAG pipeline using `R1-Distill-Llama-8B` and `R1-Distill-Qwen-7B` to handle questions as shown in Table 8.

We observe that scores get improved significantly by retrieving relevant knowledge on both LRMs, which demonstrates the lack of memorized knowledge. We would like to highlight that RAG with `R1-Distill-Qwen-7B` even surpasses closed-book `R1-Distill-Qwen-32B` on F1. This provides strong evidence that knowledge memorization is largely compromised when compressing from 32B to 7B (a substantial reduction in parameter count), whereas multihop reasoning capability is not clearly affected by this reduction. Therefore, taken together with the discussion in Section 3.3, our key finding 1 is validated.

## M ADDITIONAL DIAGNOSIS

We performed another diagnosis by running AWQ, checking if there is alignment between our interpretation results and AWQ. Since a salient weight channel typically gets scaled up more significantly than others, its scaling factor essentially serves as an importance measure that is based on activations. Removing its normalization on scaling factors that is done component by component, we extract the scaling factors from 4-bit AWQ on R1 distilled Llama-8B. We notice that the last-layer up projection gets assigned to the second largest scaling factor across all up projections by excluding the scaling factor of 1 (scaling factor of 1 means no scaling), so we see a strong alignment between AWQ and our key finding 2. The incorporation of the gradient signal in attribution patching increases our importance score of the last-layer up projection from the second largest to the largest.

## N ADDITIONAL SELECTIVE QUANTIZATION

First of all, in Table 3, both 20.0 and 6.7 are pretty low scores on AIME 2024 given that only a single component is quantized, since the original model reaches 42.2 over three runs in Table 1. We suspect a greater variability on low-score output, so we take a closer look at the outputs of quantizing 32_up and 1_up. We find that the output of only quantizing 32_up contains significantly more gibberish than 1_up, which indicates the low output quality of only quantizing 32_up. For example, the total output size of only quantizing 32_up is 10.9 MB while the total output size of only quantizing 1_up is 7.5 MB.

Taking these signals into consideration, we additionally demonstrate the lower output quality of only quantizing 32_up on AIME 2024 by performing selective 4-bit and 5-bit quantization on 32_up and 1_up. To show generalizability, we also perform selective 3-bit quantization on both components over `R1-Distill-Qwen-7B`. The resulting AIME 2024 scores are shown in Table 9. We see that only quantizing 32_up yields lower scores than 1_up on all three cases. Considering the lower average scores of only quantization 32_up in Table 3, we think the exception in Table 3 (quantizing only 1_up causes the largest drop on AIME 2024) is due to random perturbation of declined capabilities on AIME 2024.

Table 10: Analysis of test-time compute when selecting the shortest and longest 30% of responses output by each model on each benchmark. "Short" column contains performance scores of the shortest 30% of outputs from a model, while "long" column contains scores of the longest 30% of outputs. We compare the scores between "Short" and "Long" for every model and benchmark, and mark the best scores in **bold**. "Ratio" column represents the ratio of the average length (in number of tokens) of the longest 30% to that of the shortest 30%.

| Models | | AIME 2024 | | | FOLIO | | | MuSiQue | | |
|---|---|---|---|---|---|---|---|---|---|---|
| Model | Compression | Short | Long | Ratio | Short | Long | Ratio | Short | Long | Ratio |
| DeepSeek-R1 | - | **88.9** | 33.3 | 5.3 | **83.3** | 63.3 | 8.0 | **(30.0, 42.8)** | (3.3, 10.0) | 6.4 |
| DeepSeek-R1 | 2.51-bit | **100.0** | 33.3 | 4.9 | **85.0** | 71.7 | 6.5 | **(33.3, 41.9)** | (0.0, 6.9) | 7.7 |
| DeepSeek-R1 | 1.73-bit | **88.9** | 22.2 | 4.4 | **86.7** | 73.3 | 4.9 | **(30.0, 41.6)** | (10.0, 23.5) | 6.9 |
| DeepSeek-R1 | 1.58-bit | **77.8** | 44.4 | 3.8 | **80.0** | 65.0 | 5.0 | **(30.0, 41.5)** | (10.0, 16.6) | 5.9 |
| R1-Distill-Llama | Distillation | **88.9** | 11.1 | 5.4 | **80.0** | **80.0** | 4.5 | **(26.7, 38.0)** | (6.7, 18.4) | 4.0 |
| R1-Distill-Llama | Distillation & 10% sparse | **100.0** | 0.0 | 6.6 | **85.0** | 78.3 | 4.9 | **(20.0, 29.6)** | (6.7, 13.7) | 7.4 |
| R1-Distill-Llama | Distillation & 30% sparse | **88.9** | 11.1 | 5.3 | **85.0** | 76.7 | 5.0 | **(26.7, 36.9)** | (3.3, 12.5) | 8.8 |
| R1-Distill-Qwen | Distillation | **100.0** | 11.1 | 7.1 | **86.7** | 75.0 | 6.9 | **(16.7, 24.1)** | (16.7, 24.7) | 7.1 |
| R1-Distill-Qwen | Distillation & 10% sparse | **88.9** | 33.3 | 4.8 | **83.3** | 75.0 | 5.5 | **(23.3, 36.7)** | (3.3, 12.8) | 8.1 |
| R1-Distill-Qwen | Distillation & 30% sparse | **88.9** | 11.1 | 6.8 | **86.7** | 78.3 | 7.8 | **(26.7, 40.7)** | (3.3, 9.1) | 8.8 |

Table 11: Analysis of test-time compute when selecting the shortest and longest 20% of responses output by each model on each benchmark. Refer to Table 10 for the meaning of each column, except that we select shortest and longest 20% instead.

| Models | | AIME 2024 | | | FOLIO | | | MuSiQue | | |
|---|---|---|---|---|---|---|---|---|---|---|
| Model | Compression | Short | Long | Ratio | Short | Long | Ratio | Short | Long | Ratio |
| DeepSeek-R1 | - | **100.0** | 16.7 | 7.1 | **82.5** | 57.5 | 10.9 | **(30.0, 42.4)** | (5.0, 12.5) | 8.5 |
| DeepSeek-R1 | 2.51-bit | **100.0** | 33.3 | 6.6 | **87.5** | 62.5 | 8.6 | **(40.0, 50.3)** | (0.0, 4.5) | 10.7 |
| DeepSeek-R1 | 1.73-bit | **100.0** | 33.3 | 5.9 | **87.5** | 67.5 | 6.3 | **(30.0, 43.7)** | (5.0, 15.7) | 9.3 |
| DeepSeek-R1 | 1.58-bit | **100.0** | 33.3 | 5.6 | **80.0** | 72.5 | 6.2 | **(35.0, 46.3)** | (15.0, 20.0) | 7.6 |
| R1-Distill-Llama | Distillation | **83.3** | 16.7 | 7.0 | **82.5** | 75.0 | 5.7 | **(35.0, 45.2)** | (5.0, 9.2) | 5.3 |
| R1-Distill-Llama | Distillation & 10% sparse | **100.0** | 0.0 | 9.2 | **90.0** | 75.0 | 6.3 | **(20.0, 28.5)** | (5.0, 11.7) | 10.7 |
| R1-Distill-Llama | Distillation & 30% sparse | **100.0** | 16.7 | 7.1 | **90.0** | 75.0 | 6.4 | **(20.0, 31.0)** | (0.0, 7.3) | 13.0 |
| R1-Distill-Qwen | Distillation | **100.0** | 0.0 | 9.8 | **87.5** | 75.0 | 9.0 | **(20.0, 28.7)** | (20.0, 24.2) | 10.4 |
| R1-Distill-Qwen | Distillation & 10% sparse | **83.3** | 33.3 | 6.1 | **85.0** | 75.0 | 7.2 | **(30.0, 36.2)** | (0.0, 8.4) | 12.0 |
| R1-Distill-Qwen | Distillation & 30% sparse | **100.0** | 0.0 | 8.9 | **87.5** | 75.0 | 11.0 | **(30.0, 36.6)** | (5.0, 11.2) | 13.4 |

## O  TEST-TIME COMPUTE

We study the behavior of R1 and some of its compressed variants by measuring their test-time compute. Table 10 shows the analysis of test-time compute when we select the shortest and longest 30% of responses output by each model on each benchmark. We observe that shorter model outputs consistently yield better performance across three reasoning benchmarks, with only an exception on MuSiQue. Regardless of whether an LRM is compressed, if it generates significantly more tokens for a question than other questions in the same dataset, the answer is likely to be incorrect. We exclude Temporal here, because many of the compressed R1 models achieve close to 100% accuracy. The length ratios between the longest and the shortest 30% are typically greater than 4, which indicates nontrivial length differences among model outputs.

Similarly, Table 11 shows the selection of the shortest and the longest 20% of model responses. Compared to Table 10, as we move toward the extreme, the performance gap between the shortest and longest responses becomes even larger. Both R1 and its compressed variants achieve higher scores when they spend less compute during test time. After manually checking long responses, we notice that longer outputs tend to be more verbose and involve more backtracking in reasoning. This finding demonstrates the need to reduce verbosity to improve reasoning performance, which aligns with recent research (Zhang et al., 2024b).

## P  CASE STUDY

As discussed in Section 3.2, pruned models collapse on all benchmarks at certain sparsity levels. We identify a common phenomenon when a model collapses: it repeatedly generates a sentence or a

Figure 14: Two examples of the case when a model collapses and keeps repeating itself. These are two outputs (for a FOLIO question) from `DeepSeek-R1-Distill-Qwen-32B` at either 70% or 80% sparsity levels.

chunk until reaching the maximum generation length. We show two examples of this phenomenon in Figure 14. For brevity, we omit the beginning and the end of outputs.

In both examples, the pruned models are repeating themselves without pushing their reasoning chains forward, which is a signal of model collapse. When `R1-Distill-Qwen-32B` is pruned to 70% sparsity, we see that it can still organize a few sentences (*e.g.*, a chunk) to repeat. But when it is pruned to 80% sparsity, it only repeats a simple sentence. This decline of linguistic capability is common when models are pruned to high sparsities. Therefore, aggressively pruning LRMs requires careful consideration.

