# OpenReview forum: "When Reasoning Meets Compression: Understanding the Effects of LLMs Compression on Large Reasoning Models"
_ICLR.cc/2026/Conference — ICLR 2026 Poster_

### Official Review · Reviewer_G6hk · 2025-10-28

**Soundness:** 3
**Presentation:** 3
**Contribution:** 2
**Rating:** 4
**Confidence:** 3

**Summary:**

This paper investigates how compression methods including quantization, distillation, and pruning, affect the reasoning abilities of large reasoning models (LRMs).

**Strengths:**

(1) The paper is clearly written, with a well-structured presentation that is easy to follow.

(2) The motivation is articulated in a clear and convincing manner.

(3) The study provides a comprehensive analysis and offers valuable new insights.

**Weaknesses:**

(1) The chosen reasoning models are all based on DeepSeek-R1 or its distilled variants, which may constrain the generality of the claims. It is unclear whether the findings would also hold for other reasoning models such as GPT-OSS-20B or GPT-OSS-120B.

(2) The proposed layer-importance locating method is not entirely convincing. For example, Figure 2 suggests that the first-layer weights are relatively unimportant, yet Table 3 shows that quantizing the 1_up component causes the largest performance drop on AIME 2024. This apparent contradiction raises concerns about the reliability of the identified importance scores.

(3) Minor concerns: the tick labels in Figure 2 are not clearly visible, and there is a typo in line 483 (“imrpving” → “improving”).

**Questions:**

(1) Can the findings in this paper also hold for other reasoning models such as GPT-OSS-20B/120B?

(2) Why quantizing the 1_up component causes the largest performance drop on AIME 2024 but still keeps relatively good performence on other tasks.

(3) Does the identified important component also depend on the task type/task difficulties?

---

> ### Author Response · Authors · 2025-11-24
> **Response to Reviewer G6hk (Part 1)**
>
> We thank the reviewer for the helpful feedback! We plan to update our draft to incorporate the material below, addressing all reviewers’ comments before the discussion phase ends.
>
> **Weakness 1 & Question 1**: Thanks for your suggestion! Following your suggestion, we are excited to share that our findings generalize well to “Pinkstack/DistilGPT-OSS-qwen3-4B” (a distilled model from both GPT-OSS-20B and GPT-OSS-120B via supervised fine-tuning). In order to systematically investigate weight importance (such as the left part of Figure 2 and Figure 4) and quantization effect (such as Figure 3, 6, and 7), we interpret this distilled model and its 4-bit GPTQ quantized variant. We first show their one-pass benchmarking scores below. It is clear to see that MuSiQue scores are not affected by quantization, further implying that preserving weight count with lower precisions is a reasonable strategy for LRMs’ knowledge memorization (key finding 1).
>
> | Model                                         | AIME 2024 | FOLIO | Temporal | Avg  | MuSiQue     |
> |----------------------------------------------|----------:|------:|---------:|-----:|------------:|
> | DistilGPT-OSS-qwen3-4B                      |     36.7  | 82.3  |    94.4  | 71.1 | (4.0, 6.73) |
> | 4-bit GPTQ quantized DistilGPT-OSS-qwen3-4B |     26.7  | 76.4  |    86.4  | 63.2 | (4.0, 6.92) |
>
> Then, we look at our key finding 2. After running our fine-grained interpretation analysis on DistilGPT-OSS-qwen3-4B, we present its heatmaps here: https://drive.google.com/file/d/1mfrAp0UHnTyujwvsJY2LR0IlQUVn2kGj/view?usp=sharing. We see that 36_up (up_projection in the final layer) is either the most important or the second most important component across four heatmaps, which demonstrates the generalizability of our key finding 2. Moreover, whenever 36_up is ranked second on a heatmap, 36_gate is always ranked first. As both up_projection and gate_projection in the final layer are important to R1 distilled models in Figure 2 (left part) and 4, we see a high similarity of weight importance between R1 distilled models and this GPT-OSS distilled model, which indicates the possibility of finding common important components across different LRMs. On the other hand, it is worth noting that different LRMs might end up with slightly different heatmaps. For example, last-layer k_projection is highlighted in Figure 4 but appears to be less important for the GPT-OSS distilled model. This shows that our interpretation can produce customized analysis of weight importance. With more discussion offered in our paper, our key finding 2 aims to locate the import weight component that is shared by different LRMs (e.g., different model sizes and classes).
>
>
> Finally, for our key finding 3, we demonstrate its generalizability through the 4-bit GPTQ quantized DistilGPT-OSS-qwen3-4B. The heatmaps for visualizing decrease of relative importance are shown here: https://drive.google.com/file/d/1QeeKS3K5IHpF1EuTVOj2bDQIkQ1xCV4R/view?usp=sharing. Most salient decreases (outliers) come from the last-layer components across all four heatmaps, along with a few salient decreases in the second last layer. Out of a small number of outliers in a heatmap, gate_proj also has at least 2 outliers on the backtracking and uncertainty estimation capabilities. Therefore, even when we switch to a GPT-OSS distilled model, GPTQ still overly compresses the final-layer modules and MLP gate projections.
>
> We plan to add this generalizability analysis of our key findings in our revised version.

---

> > ### Author Response · Authors · 2025-11-24
> > **Response to Reviewer G6hk (Part 2)**
> >
> > **Weakness 2 & Question 2**: First of all, both 20.0 and 6.7 are pretty low scores on AIME 2024 given that only a single component is quantized, since the original model reaches 42.2 over three runs in Table 1. We suspect a greater variability on low-score output, so we take a closer look at their outputs. We find that the output of only quantizing 32_up contains significantly more gibberish than 1_up, which indicates the low output quality of only quantizing 32_up. For example, the total output size of only quantizing 32_up is 10.9 MB while the total output size of only quantizing 1_up is 7.5 MB.
> >
> > Taking these signals into consideration, we additionally demonstrate the lower output quality of only quantizing 32_up on AIME 2024 by performing selective 4-bit and 5-bit quantization on 32_up and 1_up. To show generalizability, we also perform selective 3-bit quantization on both components over R1-Distill-Qwen-7B. The resulting AIME 2024 scores are shown below. We see that only quantizing 32_up yields lower scores than 1_up on all three cases. Considering the lower average scores of only quantization 32_up in Table 3, we think the exception in Table 3 (quantizing only 1_up causes the largest drop on AIME 2024) is due to random perturbation of declined capabilities on AIME 2024.
> >
> > |                         | 32_up | 1_up |
> > |-------------------------|------:|-----:|
> > | 4-bit (R1-Distill-Llama-8B) | 20.0  | 43.3 |
> > | 5-bit (R1-Distill-Llama-8B) | 26.7  | 30.0 |
> > | 3-bit (R1-Distill-Qwen-7B)  | 16.7  | 46.7 |
> >
> >
> > **Weakness 3**: Thanks for your suggestion! We will make the heatmaps in Figure 2 larger when we are given an additional page of content. We will correct the typo in our revised draft that will be submitted later.
> >
> > **Question 3**: As long as the task is reasoning-specific and the model subject to be compressed is an LRM, our identified important component (up projection in the final layer) is expected to remain important when the task type or difficulty changes. First of all, our annotation dataset consists of 120 instances drawn from the four benchmark datasets (30 instances from each). Thus, our interpretability analysis spans four different datasets, ranging from top math challenges to multi-hop reasoning (Figure 1). Besides various reasoning types, these datasets also have different difficulty levels as explained in Section 3.1. For example, we see the largest score decrease on LRMs using R1 distilled Llama-70B as the backbone. Our comprehensive incorporation of task types and difficulties ensures the generality of our findings.
> >
> > Then, we specify in the abstract and the introduction that last-layer up projection emerges as “one of the most important” components. So there might be some other outliers that have slightly higher importance scores when the task is changed, but we expect that there will still be an outlier at the last-layer up projection. Finally, as our response to Reviewer SXNL on weakness 2, our importance measure is dependent on signals based on model activation and gradient. Since activation can be fairly stable on different inputs, we do not expect a significant change of our key finding 2 when a different reasoning task is selected.
> >
> > We will include this discussion on details of annotation dataset and generality of our findings in our revised draft.

---

### Official Review · Reviewer_ja42 · 2025-10-31

**Soundness:** 3
**Presentation:** 4
**Contribution:** 2
**Rating:** 4
**Confidence:** 3

**Summary:**

This paper investigates how compression methods affect large reasoning models (LRMs), specifically focusing on DeepSeek-R1 and its variants. The authors benchmark compressed models on four reasoning datasets and employ mechanistic interpretability techniques (difference of means and attribution patching) to identify which weights are most important for reasoning capabilities. Key findings include: (1) weight count impacts knowledge memorization more than reasoning, (2) the MLP up-projection in the final layer is critically important for distilled LRMs, and (3) current quantization methods overly compress final-layer modules and MLP gate projections.

**Strengths:**

**Comprehensive scope.** The paper systematically evaluates three major compression paradigms (quantization, distillation, and pruning) on LRMs, addressing a timely and important research gap.

**Easy to read and well-structured.** The paper maintains a clear narrative flow with consistent notation, provides context for key design choices, and connects results to takeaways, which significantly improves readability.

**Weaknesses:**

**Limited model coverage.**
The analysis centers almost exclusively on DeepSeek-R1 and its distilled variants, which makes several findings read as DeepSeek-specific behaviors rather than properties of compression that generalize across LRMs. Validating the conclusions on additional open-sourced LRM families (e.g., QwQ variants) would strengthen the generalizability claims and help disentangle model-specific effects from compression-induced phenomena.

**Imbalanced treatment across compression methods.**
The paper allocates uneven coverage across the three compression families. Distillation is represented by off-the-shelf distilled checkpoints for black-box open-source models, quantization is explored with four distinct methods, while pruning is evaluated only with SparseGPT. This asymmetry makes the comparisons look method-specific rather than family-level and may hurt perceived fairness.

**Coarse knowledge vs. reasoning disentanglement.**
The paper infers that parameter count chiefly affects knowledge based largely on lower MuSiQue EM/F1 compared to other tasks. However, this single contrast is not sufficient to attribute effects to knowledge retention versus reasoning capability. MuSiQue itself blends multi-hop reasoning with retrieval-like knowledge and is sensitive to prompt/context choices. As a result, the “parameter count affects knowledge” conclusion feels somewhat overstated. More fine-grained experiments would strengthen the claim, e.g., RAG vs. closed-book ablations across model sizes or other synthetic tasks that decouple reasoning from memorized facts.

**Minor presentation issues.** There’s a typo (“imrpving”) in Section 6.

**Questions:**

See weaknesses above.

---

> ### Author Response · Authors · 2025-11-24
> **Response to Reviewer ja42 (Part 1)**
>
> We thank the reviewer for the helpful feedback! We plan to update our draft to incorporate the material below, addressing all reviewers’ comments before the discussion phase ends.
>
> **Weakness 1 (“coverage”)**: Thanks for your suggestion! We are excited to share that our findings generalize well to “Pinkstack/DistilGPT-OSS-qwen3-4B” (a distilled model from both GPT-OSS-20B and GPT-OSS-120B via supervised fine-tuning). In order to systematically investigate weight importance (such as the left part of Figure 2 and Figure 4) and quantization effect (such as Figure 3, 6, and 7), we interpret this distilled model and its 4-bit GPTQ quantized variant. We first show their one-pass benchmarking scores below. It is clear to see that MuSiQue scores are not affected by quantization, further implying that preserving weight count with lower precisions is a reasonable strategy for LRMs’ knowledge memorization (key finding 1). We also address your concern on key finding 1 in weakness 3.
>
> | Model                                         | AIME 2024 | FOLIO | Temporal | Avg  | MuSiQue     |
> |----------------------------------------------|----------:|------:|---------:|-----:|------------:|
> | DistilGPT-OSS-qwen3-4B                      |     36.7  | 82.3  |    94.4  | 71.1 | (4.0, 6.73) |
> | 4-bit GPTQ quantized DistilGPT-OSS-qwen3-4B |     26.7  | 76.4  |    86.4  | 63.2 | (4.0, 6.92) |
>
> Then, we look at our key finding 2. After running our fine-grained interpretation analysis on DistilGPT-OSS-qwen3-4B, we present its heatmaps here: https://drive.google.com/file/d/1mfrAp0UHnTyujwvsJY2LR0IlQUVn2kGj/view?usp=sharing. We see that 36_up (up_projection in the final layer) is either the most important or the second most important component across four heatmaps, which demonstrates the generalizability of our key finding 2. Moreover, whenever 36_up is ranked second on a heatmap, 36_gate is always ranked first. As both up_projection and gate_projection in the final layer are important to R1 distilled models in Figure 2 (left part) and 4, we see a high similarity of weight importance between R1 distilled models and this GPT-OSS distilled model, which indicates the possibility of finding common important components across different LRMs. On the other hand, it is worth noting that different LRMs might end up with slightly different heatmaps. For example, last-layer k_projection is highlighted in Figure 4 but appears to be less important for the GPT-OSS distilled model. This shows that our interpretation can produce customized analysis of weight importance. With more discussion offered in our paper, our key finding 2 aims to locate the import weight component that is shared by different LRMs (e.g., different model sizes and classes).
>
> Finally, for our key finding 3, we demonstrate its generalizability through the 4-bit GPTQ quantized DistilGPT-OSS-qwen3-4B. The heatmaps for visualizing decrease of relative importance are shown here: https://drive.google.com/file/d/1QeeKS3K5IHpF1EuTVOj2bDQIkQ1xCV4R/view?usp=sharing. Most salient decreases (outliers) come from the last-layer components across all four heatmaps, along with a few salient decreases in the second last layer. Out of a small number of outliers in a heatmap, gate_proj also has at least 2 outliers on the backtracking and uncertainty estimation capabilities. Therefore, even when we switch to a GPT-OSS distilled model, GPTQ still overly compresses the final-layer modules and MLP gate projections.
>
> We plan to add this generalizability analysis of our key findings in our revised version.

---

> > ### Author Response · Authors · 2025-11-24
> > **Response to Reviewer ja42 (Part 2)**
> >
> > **Weakness 2 (“imbalanced treatment”)**: As mentioned in line 265, we require a pruned LRM that remains usable for interpretability analysis, as annotating and interpreting outputs that consist largely of gibberish would be meaningless. Therefore, in order to provide more insights on pruning, we first run additional benchmarking using a different pruning method called AlphaPruning suggested by Reviewer SXNL. Specifically, AlphaPruning computes layerwise compression ratios, and we apply SparseGPT to prune the layers accordingly. Because SparseGPT with 50% sparsity produces unusable outputs (especially on AIME 2024; see Table 1), we turn to AlphaPruning to assess whether the scores can be improved. We show AlphaPruning scores (averaged over 3-pass) on two R1 distilled models below.
> >
> > | Model                         | Compression   | AIME 2024 | FOLIO | Temporal | Avg  | MuSiQue       |
> > |------------------------------|--------------|----------:|------:|---------:|-----:|--------------:|
> > | DeepSeek-R1-Distill-Llama-70B | 50% sparsity |     26.7  | 74.2  |    97.7  | 66.2 | (5.3, 12.39)  |
> > | DeepSeek-R1-Distill-Llama-8B  | 50% sparsity |      6.7  | 61.7  |    79.6  | 49.3 | (0.0, 2.95)   |
> > | DeepSeek-R1-Distill-Llama-8B  | 30% sparsity |     41.1  | 68.9  |    82.1  | 64.1 | (0.3, 4.51)   |
> >
> >
> > First, compared to SparseGPT on the R1 distilled Llama-70B, AlphaPruning achieves higher scores on most benchmarks. However, a score of 26.7 on AIME 2024 still indicates a significant level of gibberish. We then run it on R1 distilled Llama-8B under two sparsity levels. Similarly, 50% sparsity still produces unusable outputs, so we iteratively reduce the sparsity level and ultimately settle on 30%, where much stronger reasoning performance is achieved. Although 30% sparsity (1.4x reduction in size) offers much smaller compression ratio than 4-bit quantization (4x reduction in size), its “avg” score of 64.1 is significantly higher than 4-bit GPTQ and GPTAQ. Therefore, we select AlphaPruning at 30% sparsity on the R1-distilled Llama-8B as the pruned LRM for our interpretability analysis. We show the corresponding heatmaps here: https://drive.google.com/file/d/18YnqChpo5QHXxExyBVTT3D6RinRGtii3/view?usp=sharing.
> >
> > We observe that the heatmaps of AlphaPruning at 30% appears very similar to 4-bit AWQ on R1 distilled Llama-8B (Figure 3), 4-bit GPTQ on R1 distilled Llama-8B (Figure 7), and 4-bit AWQ on R1 distilled Qwen-7B (Figure 6). Therefore, our key finding 3 can be nicely generalized to AlphaPruning as well: AlphaPruning overly compresses the final-layer modules and MLP gate projections. This additional experiment shows the effects of pruning and the generalizability of our key finding 3 on pruning methods. We will add this analysis of pruning methods in our revised draft as well.
> >
> >
> > **Weakness 3 (“disentanglement”)**: In order to further validate key finding 1, we follow your insightful suggestion and implement a RAG pipeline across two R1 distilled models with different sizes. Since all MuSiQue scores in Table 1 follow the closed-book setting as specified in line 208, we provide results from naive RAG to decouple reasoning from knowledge memorization. Specifically, we use "BAAI/bge-base-en-v1.5" as the embedding model and retrieve the top 10 chunks whose embeddings are closest to each question in MuSiQue. We run this RAG pipeline using R1-Distill-Llama-8B and R1-Distill-Qwen-7B to handle questions as shown below (the closed-book scores are copied from Table 1).
> >
> > | Setting     | LRM                  | MuSiQue      |
> > |------------|----------------------|-------------:|
> > | Closed-book| R1-Distill-Llama-8B  | (0.0, 4.43)  |
> > | RAG        | R1-Distill-Llama-8B  | (2.0, 14.52) |
> > | Closed-book| R1-Distill-Qwen-7B   | (0.0, 3.57)  |
> > | RAG        | R1-Distill-Qwen-7B   | (2.0, 13.54) |
> >
> > We observe that scores get improved significantly by retrieving relevant knowledge on both LRMs, which demonstrates the lack of memorized knowledge. We would like to highlight that RAG with R1-Distill-Qwen-7B even surpasses closed-book R1-Distill-Qwen-32B on F1. This provides strong evidence that knowledge memorization is largely compromised when compressing from 32B to 7B (a substantial reduction in parameter count), whereas multihop reasoning capability is not clearly affected by this reduction. Therefore, taken together with the discussion in Section 3.3, our key finding 1 is validated.
> >
> >
> > **Weakness 4 (“minor presentation issues”)**: Thanks for pointing it out! We were referring to “improving” and will correct it in our revised draft that will be submitted later.

---

> > > ### Comment · Reviewer_ja42 · 2025-11-28
> > >
> > > I have carefully read the authors’ response, which has addressed most of my concerns. Accordingly, I plan to adjust my overall rating.

---

### Official Review · Reviewer_SXNL · 2025-11-01

**Soundness:** 3
**Presentation:** 3
**Contribution:** 3
**Rating:** 4
**Confidence:** 3

**Summary:**

This paper systematically studies the effective of different model compression strategies on the reasoning capability of LRM, by applying reasoning-diagnosis metrics such as the steering vectors. From extensive empirical studies the authors propose certain properties and locate certain modules in LRM that are most critical to reasoning, providing insights into future designs on compression methods.

After reading the paper, I am under the impression that the paper has meaningful motivation, supported by well-designed empirical studies with strong results, and the claims are well-presented. However, the paper

1. Lacks methodological novelty and diversity in the analytical framework, using only one previously-proposed metrics. Some of the main claims of the paper, such as the localization of critical modules.
2. Could be further supported by more rigorous controlled study and evaluation metrics.

Therefore I recommend weak reject, but with room for further improvement after obtaining more insights from the authors during discussions.

**Strengths:**

1. The motivation of the paper is meaningful, as the authors focus on the effect of compression methods to (hard) reasoning tasks, which is both critical and not well-studied.
2. From comprehensive experiments the authors provides valuable insights on the design of compression methods on LRMs, that is to pay attention to certain modules important to reasoning.

**Weaknesses:**

1. The paper makes, in my opinion, a quite strong claim on localizing the reasoning ability to certain model layers. The claim that later layer is more critical to reasoning/performances is intuitive, as they have greater impact on the final output, and could be sensitive to perturbations/compressions. However the authors do not go deeper into why certain deep layers are important to reasoning.
2. For a systematic study, the authors should consider using a more diverse set of diagnosing tools, rather than only using one framework, as it would put the general applicability of the main claim under question.
3. In the paper, the locating of tokens related to certain reasoning behaviors seems not rigorous enough, as the authors only use GPT-4o to identify related tokens, which itself may have certain biases.
4. Some missing discussions on the layer/module-wise compression methods [1, 2], and identifying weights important to reasoning [3]

[1] Using Heavy-Tailed Self Regularization Theory for Improved Layer-wise Pruning of Large Language Models, https://arxiv.org/abs/2410.10912

[2] Outlier Weighed Layerwise Sparsity (OWL): A Missing Secret Sauce for Pruning LLMs to High Sparsity, https://arxiv.org/abs/2310.05175

[3] Principal Weights Emerge after Rank Reduction for Reasoning-Focused Supervised Fine-Tuning, https://arxiv.org/abs/2506.00772

**Questions:**

1. Could the authors add more discussions on recent works [1] on the principal weights of model weights that are related to reasoning performances.
2. Since the choosing of tokens related to reasoning behaviors may be critical in the determination of reasoning-related modules, I wonder whether the authors have tried using other models (other than GPT-4o) to determine the tokens. In those cases, would the overall claims still hold? If so, it will strengthen the robustness of the findings.
3. The authors proposes that certain layers/modules are more important in reasoning process, which relates to the discussion on the imbalanced quality among layers/modules. Do the authors believe that module-specific compression methods could potentially address the problem of over-compressing the important layers? There are recent works discussing module-wise compression [1, 2]
4. For empirical evidence of layer importance (Takeaway 4.1 & 4.3, Fig. 2 & 3, etc.) it would be more intuitive if the authors could present the $I_{ml}^c$ metric for the base model (before distillation) to give the reader a better idea on how the importance of each module changes, rather than only presenting the $RI_{ml}^c$.

[1] Using Heavy-Tailed Self Regularization Theory for Improved Layer-wise Pruning of Large Language Models, https://arxiv.org/abs/2410.10912

[2] Outlier Weighed Layerwise Sparsity (OWL): A Missing Secret Sauce for Pruning LLMs to High Sparsity, https://arxiv.org/abs/2310.05175

[3] Principal Weights Emerge after Rank Reduction for Reasoning-Focused Supervised Fine-Tuning, https://arxiv.org/abs/2506.00772

---

> ### Author Response · Authors · 2025-11-24
> **Response to Reviewer SXNL (Part 1)**
>
> We thank the reviewer for the helpful feedback! We plan to update our draft to incorporate the material below, addressing all reviewers’ comments before the discussion phase ends. To the best of our knowledge, we are the first to run fine-grained mechanistic interpretation on various compressed LRMs to decode the effects of compression.
>
> **Weakness 1**: Regarding “strong claim”, we are excited to share that our findings generalize well to “Pinkstack/DistilGPT-OSS-qwen3-4B” (a distilled model from both GPT-OSS-20B and GPT-OSS-120B via supervised fine-tuning). In order to systematically investigate weight importance (such as the left part of Figure 2 and Figure 4) and quantization effect (such as Figure 3, 6, and 7), we interpret this distilled model and its 4-bit GPTQ quantized variant. We first show their one-pass benchmarking scores below. It is clear to see that MuSiQue scores are not affected by quantization, further implying that preserving weight count with lower precisions is a reasonable strategy for LRMs’ knowledge memorization (key finding 1).
>
> | Model                                         | AIME 2024 | FOLIO | Temporal | Avg  | MuSiQue     |
> |----------------------------------------------|----------:|------:|---------:|-----:|------------:|
> | DistilGPT-OSS-qwen3-4B                      |     36.7  | 82.3  |    94.4  | 71.1 | (4.0, 6.73) |
> | 4-bit GPTQ quantized DistilGPT-OSS-qwen3-4B |     26.7  | 76.4  |    86.4  | 63.2 | (4.0, 6.92) |
>
> Then, we look at our key finding 2. After running our fine-grained interpretation analysis on DistilGPT-OSS-qwen3-4B, we present its heatmaps here: https://drive.google.com/file/d/1mfrAp0UHnTyujwvsJY2LR0IlQUVn2kGj/view?usp=sharing. We see that 36_up (up_projection in the final layer) is either the most important or the second most important component across four heatmaps, which demonstrates the generalizability of our key finding 2. Moreover, whenever 36_up is ranked second on a heatmap, 36_gate is always ranked first. As both up_projection and gate_projection in the final layer are important to R1 distilled models in Figure 2 (left part) and 4, we see a high similarity of weight importance between R1 distilled models and this GPT-OSS distilled model, which indicates the possibility of finding common important components across different LRMs. On the other hand, it is worth noting that different LRMs might end up with slightly different heatmaps. For example, last-layer k_projection is highlighted in Figure 4 but appears to be less important for the GPT-OSS distilled model. This shows that our interpretation can produce customized analysis of weight importance. With more discussion offered in our paper, our key finding 2 aims to locate the import weight component that is shared by different LRMs (e.g., different model sizes and classes).
>
> Finally, for our key finding 3, we demonstrate its generalizability through the 4-bit GPTQ quantized DistilGPT-OSS-qwen3-4B. The heatmaps for visualizing decrease of relative importance are shown here: https://drive.google.com/file/d/1QeeKS3K5IHpF1EuTVOj2bDQIkQ1xCV4R/view?usp=sharing. Most salient decreases (outliers) come from the last-layer components across all four heatmaps, along with a few salient decreases in the second last layer. Out of a small number of outliers in a heatmap, gate_proj also has at least 2 outliers on the backtracking and uncertainty estimation capabilities. Therefore, even when we switch to a GPT-OSS distilled model, GPTQ still overly compresses the final-layer modules and MLP gate projections.
>
> Drawing connections to equations in Sections 2.2 and 2.3, we address your concern on “not go deeper”  from the perspectives of our interpretation framework and distillation effect. In the next weakness point, we use AWQ as an additional diagnosis tool to further validate our findings.
>
> When we compute the steering vector via difference of means, we are essentially extracting the deep representations of the target reasoning behavior based on activations. Moving on to attribution patching, we additionally incorporate the gradient with respect to the cross-entropy loss. Therefore, our measure of casual relationship (importance scores) depends heavily on activation and first-order (gradient) signals. When we claim certain components are important in our paper (e.g., final-layer up projection), that means they end up with relatively higher magnitude of activations and gradients.
>
> We would also like to highlight Section 4.3, where we investigate why the final-layer up projection is important for reasoning. Combining the discussion in Question 4 below, we observe that the base Llama-3.1-8B prior to distillation plays little role in shaping the reasoning capabilities of the R1 distilled Llama-8B (LRM). The importance of final-layer up projection increases significantly during distillation with SFT, so this distillation effect explains why this component becomes one of the most important, as specified in Takeaway 4.3.

---

> ### Author Response · Authors · 2025-11-24
> **Response to Reviewer SXNL (Part 2)**
>
> **Weakness 2**: Thanks for your suggestion! We performed another diagnosis by running AWQ, checking if there is alignment between our interpretation results and AWQ. Since a salient weight channel typically gets scaled up more significantly than others, its scaling factor essentially serves as an importance measure that is based on activations. Removing its normalization on scaling factors that is done component by component, we extract the scaling factors from 4-bit AWQ on R1 distilled Llama-8B. We notice that the last-layer up projection gets assigned to the second largest scaling factor across all up projections by excluding the scaling factor of 1 (scaling factor of 1 means no scaling), so we see a strong alignment between AWQ and our key finding 2. The incorporation of the gradient signal as discussed above increases our importance score of the last-layer up projection from the second largest to the largest.
>
> **Weakness 3 and Question 2**: Thanks for your suggestion! It is important to find a validated approach to automatically perform annotation, as a large number of $s_i^c$ would help the computation of importance scores. We recently have tried GPT-5 on a small number of instances and do not notice a clear difference. Putting the outputs of GPT-4o and GPT-5 (under the same temperature) side by side on the same instance, we could not tell which output is produced by GPT-5. Thus, we find GPT-4o is capable enough to handle our annotation task.
>
> As mentioned in line 141, we closely followed Venhoff et al. to define our target reasoning behaviors and adopt GPT-4o for annotating token sequences of each behavior. Therefore, we did not change the behavior definitions and instruction prompts. The most salient adjustable parameter is the temperature, and we used the default temperature value (1) in our annotation pipeline. Following your suggestion, we study whether the change of temperature would overturn our annotation results.
>
> We choose a low temperature value (0.1) and see whether more deterministic outputs would affect our annotation results significantly. Setting temperature to 0.1, we first use GPT-4o with the exact same behavior definitions and instruction prompts to collect annotation. Then, we perform human validation to judge the output quality and stability when a lower temperature is set. To score a model output instance on a specific behavior, we instruct the evaluator to go through all the behavior tokens that are tagged and compute the percentage of correctly tagged behavior. For example, suppose GPT-4o tags 20 “backtracking” token sequences in an output instance and the evaluator finds 18 of them closely following the definition of “backtracking”, the score of this instance is 0.9. There are cases when an instance ends up with no tagged behavior. Since we encourage a conservative annotation style (the value of $s_i^c$ is large enough for computation such as 1000, so it is fine if GPT-4o misses some token sequences of the target behavior), we assign the score of 1 (maximum score) to those instances ending up with 0 tagged behavior. For each behavior, the evaluator samples 10 instances out of 120 (with replacement) and scores them. We present the average score of 10 sampled instances under 2 temperatures and the Spearman $\rho$ with 95% confidence interval below.
>
> | Behavior              | Avg (temp=1) | Avg (temp=0.1) | Spearman $\rho$ | CI               |
> |-----------------------|----------------------:|------------------------:|----------------:|------------------|
> | Backtracking          |                  0.90 |                    0.94 |           0.892 | [0.474, 1.000]   |
> | Uncertainty Estimation|                  0.78 |                    0.75 |           0.803 | [0.416, 1.000]   |
> | Example Testing       |                  0.90 |                    0.87 |           0.862 | [0.645, 1.000]   |
> | Adding Knowledge      |                  0.86 |                    0.96 |           0.486 | [-0.218, 1.000]  |
>
> As shown in the table, the average rating scores of both temperatures are high across 4 behaviors, which indicates reliable annotation results done by GPT-4o. Comparing the results of both temperatures side-by-side, we observe that temperature = 1 tends to produce more nested and unclosed token sequences, which is expected due to higher randomness. This does not affect our interpretation analysis, since we exclude those nested or unclosed sequences.
>
> As for Spearman $\rho$, we see that the two rating scores on all 4 reasoning behaviors exhibit at least a moderate and often strong correlation, which demonstrates stable rank ordering. For “adding knowledge” behavior, the 95% confidence interval is wide, indicating uncertainty due to the small sample size of 10. Given high average rating scores, we conclude using the default temperature value does not significantly affect the correctness of annotation. Therefore, the robustness of our findings is validated.

---

> > ### Author Response · Authors · 2025-11-24
> > **Response to Reviewer SXNL (Part 3)**
> >
> > **Weakness 4, Question 1, and 3**: Thanks for bringing up these insightful papers! Compared to these works, our paper decodes the effects of compression on LRMs and validates the usefulness and generalizability of fine-grained mechanistic interpretation for LRM compression. In [3], “weights with the largest magnitude after low-rank approximation” are considered important, which is a useful signal for sparse fine-tuning. Similarly, for weight-only compression, our paper proposes to adopt signals from mechanistic interpretation, AWQ claims to adopt the activation signals, and the second-order information is used in SparseGPT and GPTQ for importance measure. From a rigorous perspective, it is important to first check whether magnitude after low-rank approximation is indeed a useful signal for weight-only compression, since sparse training and compression may be different. Efforts need to be paid to apply this signal on weight-only compression, so benchmarking would be helpful such as our performance benchmarking on various compression methods. It is valuable to keep looking for novel importance signals. After that, we envision that a successful compression method might need to incorporate multiple signals or merge them, as some may benefit from a change in the compression goal (e.g., changing the domain or the compression ratio).
> >
> > [1] and [2] are very representative module-specific compression methods, and we think this research direction could potentially address the problem of over-compressing the important weights. This is also one of our motivations of locating important weights. As a way of enriching module-specific compression, we speculate that using non-uniform compression ratios across different modules (e.g., unstructured pruning and mix-precision quantization) might pose challenges for inference speedup. A potential solution of being more hardware-efficient is to enforce uniform ratios with protection mechanisms on important weights. For example, AWQ protects salient weights identified via activation signals by scaling them up, while still using a uniform compression ratio across modules.
> >
> > We will include this discussion in our revised draft.
> >
> > [1] Using Heavy-Tailed Self Regularization Theory for Improved Layer-wise Pruning of Large Language Models, https://arxiv.org/abs/2410.10912
> >
> > [2] Outlier Weighed Layerwise Sparsity (OWL): A Missing Secret Sauce for Pruning LLMs to High Sparsity, https://arxiv.org/abs/2310.05175
> >
> > [3] Principal Weights Emerge after Rank Reduction for Reasoning-Focused Supervised Fine-Tuning, https://arxiv.org/abs/2506.00772
> >
> > **Question 4**: Thanks for your suggestion! Complementing the right part (change of relative importance from R1-Distill-Llama-8B to its base model before distillation) of Figure 2, we present the importance scores of the base model (Llama-3.1-8B) here: https://drive.google.com/file/d/1zqiHa_ytmit3uhv93tpGcAXcZ8CLTpFs/view?usp=sharing. As we can see, the heatmaps of the base model look very different from those of the R1-distilled model (left part of Figure 2). There are many differences. For example, compared to uncertainty estimation behavior, other reasoning behaviors do not present a clear outlier of importance, which indicates more evenly distributed causal relationships across weight matrices on backtracking, example testing, and adding knowledge. As for uncertainty estimation, the last-layer o_projection has the highest importance, whereas the up projection in the final layer is not even ranked among the top three components of that layer. Therefore, we could reach a completely different conclusion from key finding 2 if we were to interpret the base model rather than the LRMs. The commonality between the two parts of Figure 2, together with the difference between the base model and the right part of Figure 2, demonstrates Takeaway 4.3 in our paper (“important weights of the R1 distilled models are mainly the result of the distillation effect”). This discussion complements Section 4.3, and we will include it in our revised draft.

---

> > > ### Comment · Reviewer_SXNL · 2025-11-28
> > > **Thank you for addressing my concerns.**
> > >
> > > I appreciate the authors for their efforts and for providing complementary results and findings that make the overall findings more generalized and comprehensive. All my concerns are now addressed. I encourage the authors to incorporate the new results in the revised draft.
> > >
> > > In light of these improvements, I will raise my evaluation of this paper.

---

### Official Review · Reviewer_PAMX · 2025-11-01

**Soundness:** 3
**Presentation:** 3
**Contribution:** 2
**Rating:** 4
**Confidence:** 4

**Summary:**

This paper presents a two-fold analysis of the effects of compression on Large Reasoning Models (LRMs), using DeepSeek-R1 as its primary case study. First, it provides a comprehensive benchmark of three major compression families—quantization, distillation, and pruning—evaluating their impact on performance across a diverse set of reasoning datasets (AIME 2024, FOLIO, Temporal Sequences, and MuSiQue). Second, and more notably, the paper employs mechanistic interpretability techniques (adapting difference of means and attribution patching) to identify which specific model components are causally important for reasoning. The authors find that current compression methods, particularly quantization, disproportionately harm these critical components. They validate this finding by showing that selectively protecting these components (e.g., the final-layer MLP) during quantization significantly recovers lost performance.

**Strengths:**

Novelty of Interpretation: The paper's primary strength is its application of mechanistic interpretability to the problem of model compression. It moves beyond standard "accuracy vs. bits/size" tables to provide a causal analysis of why and where performance degrades, which is a valuable contribution.

Actionable Findings: The analysis yields clear, actionable insights. The identification of the final-layer mlp.up_proj as a critical component for reasoning (in R1-distilled models) and the finding that popular quantization methods (AWQ, GPTQ) overly compress these final layers are important discoveries.

Strong Validation: The experiment in Section 5.2 is compelling. Demonstrating that protecting just 2% of weights (the final-layer MLPs) from quantization can recover 6.57% in average accuracy on a 3-bit model provides strong validation for the paper's entire interpretability pipeline.

**Weaknesses:**

Generalizability: The analysis is heavily centered on DeepSeek-R1 and its specific distilled variants (R1-Distill-Llama, R1-Distill-Qwen). It is unclear if the central findings (e.g., the high importance of the final-layer mlp.up_proj) are a general feature of all LRMs or an artifact of the specific distillation-with-SFT process used to create the R1 models.

Subjectivity in Methodology: The interpretability analysis (Section 2.2) relies on prompting GPT-4o to locate token sequences corresponding to four specific reasoning behaviors. This labeling process seems subjective and could introduce noise or bias. The robustness of the resulting steering vectors is highly dependent on the quality of this heuristic.

Underdeveloped Pruning Analysis: While pruning is introduced as one of the three main compression methods, it is quickly dismissed after benchmarking shows it performs poorly (e.t., at 50% sparsity). The subsequent mechanistic analysis focuses almost exclusively on distillation and quantization, making the pruning aspect of the paper feel incomplete.

**Questions:**

1. Can the authors comment on whether the "final-layer importance" finding is specific to the R1-distillation process? Have you tried applying your interpretability analysis to a standard, non-distilled model (e.g., base Llama 3) that has been fine-tuned for reasoning? Would you expect to see the same components identified as critical?

2. Could you elaborate on the validation process for the GPT-4o labeling of reasoning behaviors? How sensitive are the final importance scores (and the resulting conclusions) to potential inaccuracies or inconsistencies in this automated labeling process?

3. The paper notes (Takeaway 3.3) that pruning/distillation (reducing parameter count) hurts knowledge memorization (MuSiQue) more severely than reasoning (AIME, FOLIO). Quantization (reducing precision) seems to have a less detrimental effect on knowledge. Could you expand on this distinction? Why do you hypothesize that parametric knowledge is so much more sensitive to parameter count than to parameter precision?

I would like to improve my scores if authors can solve my questions.

---

> ### Author Response · Authors · 2025-11-24
> **Response to Reviewer PAMX (Part 1)**
>
> We thank the reviewer for the helpful feedback! We plan to update our draft to incorporate the material below, addressing all reviewers’ comments before the discussion phase ends.
>
> **Weakness 1 (“generalizability”)**: Thanks for your suggestion! Following your suggestion, we are excited to share that our findings generalize well to “Pinkstack/DistilGPT-OSS-qwen3-4B” (a distilled model from both GPT-OSS-20B and GPT-OSS-120B via supervised fine-tuning). We run our interpretation on distilled models, since they are among the most popular LRMs under 8B in size. In order to systematically investigate weight importance (such as the left part of Figure 2 and Figure 4) and quantization effect (such as Figure 3, 6, and 7), we interpret this distilled model and its 4-bit GPTQ quantized variant. We first show their one-pass benchmarking scores below. It is clear to see that MuSiQue scores are not affected by quantization, further implying that preserving weight count with lower precisions is a reasonable strategy for LRMs’ knowledge memorization (key finding 1).
>
> | Model                                         | AIME 2024 | FOLIO | Temporal | Avg  | MuSiQue     |
> |----------------------------------------------|----------:|------:|---------:|-----:|------------:|
> | DistilGPT-OSS-qwen3-4B                      |     36.7  | 82.3  |    94.4  | 71.1 | (4.0, 6.73) |
> | 4-bit GPTQ quantized DistilGPT-OSS-qwen3-4B |     26.7  | 76.4  |    86.4  | 63.2 | (4.0, 6.92) |
>
> Then, we look at our key finding 2. After running our fine-grained interpretation analysis on DistilGPT-OSS-qwen3-4B, we present its heatmaps here: https://drive.google.com/file/d/1mfrAp0UHnTyujwvsJY2LR0IlQUVn2kGj/view?usp=sharing. We see that 36_up (up_projection in the final layer) is either the most important or the second most important component across four heatmaps, which demonstrates the generalizability of our key finding 2. Moreover, whenever 36_up is ranked second on a heatmap, 36_gate is always ranked first. As both up_projection and gate_projection in the final layer are important to R1 distilled models in Figure 2 (left part) and 4, we see a high similarity of weight importance between R1 distilled models and this GPT-OSS distilled model, which indicates the possibility of finding common important components across different LRMs. On the other hand, it is worth noting that different LRMs might end up with slightly different heatmaps. For example, last-layer k_projection is highlighted in Figure 4 but appears to be less important for the GPT-OSS distilled model. This shows that our interpretation can produce customized analysis of weight importance. With more discussion offered in our paper, our key finding 2 aims to locate the import weight component that is shared by different LRMs (e.g., different model sizes and classes).
>
> Finally, for our key finding 3, we demonstrate its generalizability through the 4-bit GPTQ quantized DistilGPT-OSS-qwen3-4B. The heatmaps for visualizing decrease of relative importance are shown here: https://drive.google.com/file/d/1QeeKS3K5IHpF1EuTVOj2bDQIkQ1xCV4R/view?usp=sharing. Most salient decreases (outliers) come from the last-layer components across all four heatmaps, along with a few salient decreases in the second last layer. Out of a small number of outliers in a heatmap, gate_proj also has at least 2 outliers on the backtracking and uncertainty estimation capabilities. Therefore, even when we switch to a GPT-OSS distilled model, GPTQ still overly compresses the final-layer modules and MLP gate projections.
>
> We plan to add this generalizability analysis of our key findings in our revised version.

---

> > ### Author Response · Authors · 2025-11-24
> > **Response to Reviewer PAMX (Part 2)**
> >
> > **Weakness 2 (“subjectivity”) and Question 2**: It is important to find a validated approach to automatically perform annotation, as a large number of $s_i^c$ would help the computation of importance scores. As mentioned in line 141, we closely followed Venhoff et al. to define our target reasoning behaviors and adopt GPT-4o for annotating token sequences of each behavior. Therefore, we did not change the behavior definitions and instruction prompts. The most salient adjustable parameter is the temperature, and we used the default temperature value (1) in our annotation pipeline. We study whether the change of temperature would overturn our annotation results.
> >
> > We choose a low temperature value (0.1) and see whether more deterministic outputs would affect our annotation results significantly. Setting temperature to 0.1, we first use GPT-4o with the exact same behavior definitions and instruction prompts to collect annotation. Then, we perform human validation to judge the output quality and stability when a lower temperature is set. To score a model output instance on a specific behavior, we instruct the evaluator to go through all the behavior tokens that are tagged and compute the percentage of correctly tagged behavior. For example, suppose GPT-4o tags 20 “backtracking” token sequences in an output instance and the evaluator finds 18 of them closely following the definition of “backtracking”, the score of this instance is 0.9. There are cases when an instance ends up with no tagged behavior. Since we encourage a conservative annotation style (the value of $s_i^c$ is large enough for computation such as 1000, so it is fine if GPT-4o misses some token sequences of the target behavior), we assign the score of 1 (maximum score) to those instances ending up with 0 tagged behavior. For each behavior, the evaluator samples 10 instances out of 120 (with replacement) and scores them. We present the average score of 10 sampled instances under 2 temperatures and the Spearman $\rho$ with 95% confidence interval below.
> >
> > | Behavior              | Avg (temp=1) | Avg (temp=0.1) | Spearman $\rho$ | CI               |
> > |-----------------------|----------------------:|------------------------:|----------------:|------------------|
> > | Backtracking          |                  0.90 |                    0.94 |           0.892 | [0.474, 1.000]   |
> > | Uncertainty Estimation|                  0.78 |                    0.75 |           0.803 | [0.416, 1.000]   |
> > | Example Testing       |                  0.90 |                    0.87 |           0.862 | [0.645, 1.000]   |
> > | Adding Knowledge      |                  0.86 |                    0.96 |           0.486 | [-0.218, 1.000]  |
> >
> > As shown in the table, the average rating scores of both temperatures are high across 4 behaviors, which indicates reliable annotation results done by GPT-4o. Therefore, our final importance scores are not sensitive to potential inaccuracies of GPT-4o. Comparing the results of both temperatures side-by-side, we observe that temperature = 1 tends to produce more nested and unclosed token sequences, which is expected due to higher randomness. This does not affect our interpretation analysis, since we exclude those nested or unclosed sequences.
> >
> > As for Spearman $\rho$, we see that the two rating scores on all 4 reasoning behaviors exhibit at least a moderate and often strong correlation, which demonstrates stable rank ordering. For “adding knowledge” behavior, the 95% confidence interval is wide, indicating uncertainty due to the small sample size of 10. Given high average rating scores, we conclude using the default temperature value does not significantly affect the correctness of annotation. Therefore, the robustness of our findings is validated.

---

> > > ### Author Response · Authors · 2025-11-24
> > > **Response to Reviewer PAMX (Part 3)**
> > >
> > > **Weakness 3 (“pruning analysis”)**: As mentioned in line 265, we require a pruned LRM that remains usable for interpretability analysis, as annotating and interpreting outputs that consist largely of gibberish would be meaningless. Therefore, in order to provide more insights on pruning, we first run additional benchmarking using a different pruning method called AlphaPruning suggested by Reviewer SXNL. Specifically, AlphaPruning computes layerwise compression ratios, and we apply SparseGPT to prune the layers accordingly. Because SparseGPT with 50% sparsity produces unusable outputs (especially on AIME 2024; see Table 1), we turn to AlphaPruning to assess whether the scores can be improved. We show AlphaPruning scores (averaged over 3-pass) on two R1 distilled models below.
> > >
> > > | Model                         | Compression   | AIME 2024 | FOLIO | Temporal | Avg  | MuSiQue       |
> > > |------------------------------|--------------|----------:|------:|---------:|-----:|--------------:|
> > > | DeepSeek-R1-Distill-Llama-70B | 50% sparsity |     26.7  | 74.2  |    97.7  | 66.2 | (5.3, 12.39)  |
> > > | DeepSeek-R1-Distill-Llama-8B  | 50% sparsity |      6.7  | 61.7  |    79.6  | 49.3 | (0.0, 2.95)   |
> > > | DeepSeek-R1-Distill-Llama-8B  | 30% sparsity |     41.1  | 68.9  |    82.1  | 64.1 | (0.3, 4.51)   |
> > >
> > > First, compared to SparseGPT on the R1 distilled Llama-70B, AlphaPruning achieves higher scores on most benchmarks. However, a score of 26.7 on AIME 2024 still indicates a significant level of gibberish. We then run it on R1 distilled Llama-8B under two sparsity levels. Similarly, 50% sparsity still produces unusable outputs, so we iteratively reduce the sparsity level and ultimately settle on 30%, where much stronger reasoning performance is achieved. Although 30% sparsity (1.4x reduction in size) offers much smaller compression ratio than 4-bit quantization (4x reduction in size), its “avg” score of 64.1 is significantly higher than 4-bit GPTQ and GPTAQ. Therefore, we select AlphaPruning at 30% sparsity on the R1-distilled Llama-8B as the pruned LRM for our interpretability analysis. We show the corresponding heatmaps here: https://drive.google.com/file/d/18YnqChpo5QHXxExyBVTT3D6RinRGtii3/view?usp=sharing.
> > >
> > > We observe that the heatmaps of AlphaPruning at 30% appears very similar to 4-bit AWQ on R1 distilled Llama-8B (Figure 3), 4-bit GPTQ on R1 distilled Llama-8B (Figure 7), and 4-bit AWQ on R1 distilled Qwen-7B (Figure 6). Therefore, our key finding 3 can be nicely generalized to AlphaPruning as well: AlphaPruning overly compresses the final-layer modules and MLP gate projections. This additional experiment shows the effects of pruning and the generalizability of our key finding 3 on pruning methods. We will add this analysis of pruning methods in our revised draft as well.
> > >
> > > **Question 1**: As we have shown in weakness 1 (our interpretation on DistilGPT-OSS-qwen3-4B for key finding 2), our key finding 2 is not specific to the R1 distillation process. This finding also generalizes to distillation done on GPT-OSS-20B and GPT-OSS-120B, which are non-R1 LRMs.
> > >
> > > We have interpreted the base model as well, since that is how we obtain the right part of Figure 2. Complementing the right part (change of relative importance from R1-Distill-Llama-8B to its base model before distillation) of Figure 2, we present the importance scores of the base model (Llama-3.1-8B) here: https://drive.google.com/file/d/1zqiHa_ytmit3uhv93tpGcAXcZ8CLTpFs/view?usp=sharing. As we can see, the heatmaps of the base model look very different from those of the R1-distilled model (left part of Figure 2). There are many differences. For example, compared to uncertainty estimation behavior, other reasoning behaviors do not present a clear outlier of importance, which indicates more evenly distributed causal relationships across weight matrices on backtracking, example testing, and adding knowledge. As for uncertainty estimation, the last-layer o_projection has the highest importance, whereas the up projection in the final layer is not even ranked among the top three components of that layer. Therefore, we could reach a completely different conclusion from key finding 2 if we were to interpret the base model rather than the LRMs. The commonality between the two parts of Figure 2, together with the difference between the base model and the right part of Figure 2, demonstrates Takeaway 4.3 in our paper (“important weights of the R1 distilled models are mainly the result of the distillation effect”). This discussion complements Section 4.3, and we will include it in our revised draft.

---

> > > > ### Author Response · Authors · 2025-11-24
> > > > **Response to Reviewer PAMX (Part 4)**
> > > >
> > > > **Question 3**: Our discussion in Section 3.3 concentrates on the significantly lower MuSiQue scores as we move from R1 to different compressed variants. We hypothesize that parametric knowledge is more sensitive to parameter count, because pruned R1 distilled Llama-70B in Table 2 collapses on MuSiQue even earlier than AIME 2024 as explained in line 347. Other notable phenomena include the slightly stronger reasoning performance of R1 distilled Qwen-32B than Llama-70B, but with collapsed MuSiQue scores at even 0% sparsity, as explained in line 323.
> > > >
> > > > In order to further validate our key finding 1, we implement a RAG pipeline across two R1 distilled models with different sizes. Since all MuSiQue scores in Table 1 follow the closed-book setting, we provide results from naive RAG to decouple reasoning from knowledge memorization. Specifically, we use "BAAI/bge-base-en-v1.5" as the embedding model and retrieve the top 10 chunks whose embeddings are closest to each question in MuSiQue. We run this RAG pipeline using R1-Distill-Llama-8B and R1-Distill-Qwen-7B to handle questions as shown below (the closed-book scores are copied from Table 1).
> > > >
> > > > | Setting     | LRM                  | MuSiQue      |
> > > > |------------|----------------------|-------------:|
> > > > | Closed-book| R1-Distill-Llama-8B  | (0.0, 4.43)  |
> > > > | RAG        | R1-Distill-Llama-8B  | (2.0, 14.52) |
> > > > | Closed-book| R1-Distill-Qwen-7B   | (0.0, 3.57)  |
> > > > | RAG        | R1-Distill-Qwen-7B   | (2.0, 13.54) |
> > > >
> > > > We observe that scores get improved significantly by retrieving relevant knowledge on both LRMs, which demonstrates the lack of memorized knowledge. We would like to highlight that RAG with R1-Distill-Qwen-7B even surpasses closed-book R1-Distill-Qwen-32B on F1. This provides strong evidence that knowledge memorization is largely compromised when compressing from 32B to 7B (a substantial reduction in parameter count), whereas multihop reasoning capability is not clearly affected by this reduction. Therefore, taken together with the discussion in Section 3.3, our key finding 1 is validated. We will expand Section 3.3 with more details in our revised draft.

---

### Official Review · Reviewer_ST2S · 2025-11-02

**Soundness:** 4
**Presentation:** 3
**Contribution:** 3
**Rating:** 6
**Confidence:** 4

**Summary:**

This paper studies how compression (quantization, distillation, pruning) impacts reasoning in LRMs (DeepSeek-R1 and distilled Llama/Qwen) via (i) a broad benchmark on AIME-2024, FOLIO, Temporal Sequences, and MuSiQue, and (ii) mechanistic analyses that compute behavior-specific importance for every linear module using difference-of-means and attribution-patching. Key claims include: 2.51-bit dynamic quantization of R1 attains near-R1 performance; the final-layer MLP up-projection is consistently most reasoning-critical; and protecting ~2% of weights (final-layer MLPs) recovers +6.57% average accuracy for 3-bit AWQ.

**Strengths:**

1、 Comprehensive scope across three compression families with clear head-to-head tables and multiple distilled sizes (70B/32B/8B/7B).

2、Fine-grained mechanistic lens (module-level DoM + attribution-patching) tied to four reasoning behaviors (backtracking, uncertainty estimation, example testing, adding knowledge).

3、 Actionable insight: the final-layer up-projection is most critical; selectively quantizing it alone (≈0.7% of weights) causes a large drop, while protecting final-layer MLPs at 16-bit raises a 3-bit model by +6.57% on average.

4、 Useful observations on collapse points (e.g., pruning ≥50% collapses; 3-bit baselines stress on harder tasks) and knowledge vs. reasoning separation (MuSiQue).

**Weaknesses:**

1、Robustness & Statistics. Behavior labeling robustness is under-specified (prompt/threshold/seed); key results lack rank-stability and uncertainty (variance, 95% CIs, significance).

2、Metric & Visualization Choice. RI plots zero out increases, risking masked compensations; provide justified alternatives (report both ↑/↓ and net change).

3、Coverage & Generalization. Pruning analysis is shallow (no mechanistic view beyond collapse); external validity is limited (mainly R1-distilled 8B/7B, few non-R1 families).

**Questions:**

1) Behavior-label robustness.
Vary the behavior taxonomy/prompt/thresholds and report rank stability (Kendall’s τ / Spearman ρ) with 95% CIs; include a leave-one-behavior-out analysis.
2) RI visualization choice.
Justify zeroing positive RI deltas and provide an alternative view that reports both increases and decreases, plus a net-change summary.
3) Selective protection trade-offs. Provide an accuracy vs. protected-ratio (0.5–5%) curve and the latency/memory overhead; compare protecting final-layer gate vs. up vs. both.
4) 2.51-bit dynamic quantization details.
Specify the skip policy (which layers stay high-precision), calibration set, and per-module bit-allocation histograms; compare to AWQ/GPTQ under identical calibration on quantization error, attribution-preservation, and end-task metrics.

---

> ### Author Response · Authors · 2025-11-24
> **Response to Reviewer ST2S (Part 1)**
>
> We thank the reviewer for the helpful feedback! We plan to update our draft to incorporate the material below, addressing all reviewers’ comments before the discussion phase ends. To the best of our knowledge, we are the first to run fine-grained mechanistic interpretation on various compressed LRMs to decode the effects of compression.
>
> **Weakness 1 and Question 1**: Thanks for your detailed suggestion! It is important to find a validated approach to automatically perform annotation, as a large number of $s_i^c$ would help the computation of importance scores. As mentioned in line 141, we closely followed Venhoff et al. to define our target reasoning behaviors and adopt GPT-4o for annotating token sequences of each behavior. Therefore, we did not change the behavior definitions and instruction prompts. The most salient adjustable parameter is the temperature, and we used the default temperature value (1) in our annotation pipeline. Following your suggestion, we study whether the change of temperature would overturn our annotation results.
>
> We choose a low temperature value (0.1) and see whether more deterministic outputs would affect our annotation results significantly. Setting temperature to 0.1, we first use GPT-4o with the exact same behavior definitions and instruction prompts to collect annotation. Then, we perform human validation to judge the output quality and stability when a lower temperature is set. To score a model output instance on a specific behavior, we instruct the evaluator to go through all the behavior tokens that are tagged and compute the percentage of correctly tagged behavior. For example, suppose GPT-4o tags 20 “backtracking” token sequences in an output instance and the evaluator finds 18 of them closely following the definition of “backtracking”, the score of this instance is 0.9. There are cases when an instance ends up with no tagged behavior. Since we encourage a conservative annotation style (the value of $s_i^c$ is large enough for computation such as 1000, so it is fine if GPT-4o misses some token sequences of the target behavior), we assign the score of 1 (maximum score) to those instances ending up with 0 tagged behavior. For each behavior, the evaluator samples 10 instances out of 120 (with replacement) and scores them. We present the average score of 10 sampled instances under 2 temperatures and the Spearman $\rho$ with 95% confidence interval below.
>
> | Behavior              | Avg (temp=1) | Avg (temp=0.1) | Spearman $\rho$ | CI               |
> |-----------------------|----------------------:|------------------------:|----------------:|------------------|
> | Backtracking          |                  0.90 |                    0.94 |           0.892 | [0.474, 1.000]   |
> | Uncertainty Estimation|                  0.78 |                    0.75 |           0.803 | [0.416, 1.000]   |
> | Example Testing       |                  0.90 |                    0.87 |           0.862 | [0.645, 1.000]   |
> | Adding Knowledge      |                  0.86 |                    0.96 |           0.486 | [-0.218, 1.000]  |
>
> As shown in the table, the average rating scores of both temperatures are high across 4 behaviors, which indicates reliable annotation results done by GPT-4o. Comparing the results of both temperatures side-by-side, we observe that temperature = 1 tends to produce more nested and unclosed token sequences, which is expected due to higher randomness. This does not affect our interpretation analysis, since we exclude those nested or unclosed sequences.
>
> As for Spearman $\rho$, we see that the two rating scores on all 4 reasoning behaviors exhibit at least a moderate and often strong correlation, which demonstrates stable rank ordering. For “adding knowledge” behavior, the 95% confidence interval is wide, indicating uncertainty due to the small sample size of 10. Given high average rating scores, we conclude using the default temperature value does not significantly affect the correctness of annotation.
>
> We also would like to emphasize that most scores of our benchmarking results in Table 1 are averaged over 3 passes (line 216), so we can provide their variance. Since applying SparseGPT with 50% sparsity yields significantly lower scores (a 2x reduction in size for pruning versus 4x reduction for 4-bit quantization), we regard the score variability as a minor concern and therefore did not run interpretability analysis on the pruned models, as discussed in line 265. To the best of our knowledge, we present the first benchmarking work that systematically compares quantization and pruning scores on LRMs and shows that even unstructured pruning (typically unstructured pruning yields the better accuracy than structured pruning) with much lower compression ratio may yield worse reasoning accuracy than quantization. Further studying the effect of pruning, we have additionally pruned an LRM to a lower sparsity (30%) and interpreted the pruned model in weakness 3.

---

> > ### Author Response · Authors · 2025-11-24
> > **Response to Reviewer ST2S (Part 2)**
> >
> > **Weakness 2 and Question 2**: Thanks for your suggestion! We have a dedicated paragraph (the second paragraph in Section 2.3) to explain why we only visualize the decreases in relative importance. First, we track changes in relative importance from a more reasoning-capable model to a less reasoning-capable model (rather than the other way around). Then, we mention to only consider decreases to show our interest in tracking the cases “when the reasoning capability of a weight matrix is diminished”. In other words, when an LRM is compressed via suboptimal pruning or quantization, salient weight components are more likely to show decreased rather than increased relative importance, because suboptimal compression tends to preserve or harm, but not improve, the reasoning capabilities of important components. For compression methods that fully preserve or increase the relative importance of some salient components, tracking these increases adds little value, as it essentially confirms that the methods are performing well. It is more valuable to visualize the effects of compression with an emphasis on the weakness of current compression methods, which can inspire future improvement such as our key finding 3.
> >
> > On the other hand, in order to show all aspects of importance shift suggested, we draw the heatmaps of only considering increases (https://drive.google.com/file/d/1C2TWBIffUJZ1fl1NjFmHQ47YcAUKXJw9/view?usp=sharing) and those that reflect the net change (https://drive.google.com/file/d/1wqm_r4-2qzbc46XvGdE9Lu4HoKTHTPvq/view?usp=sharing) from R1 distilled Llama-8B to its 4-bit AWQ variant. We see that the net-change heatmaps appear quite similar to those that only consider decreases (Figure 3). Specifically, the net-change heatmaps primarily highlight only a few components from the increase-only heatmaps based on Figure 3, and our key finding 3 still holds (researchers are recommended to prioritize the preservation of final-layer-module importance in order to improve current quantization methods). The similarity between Figure 3 and net-change heatmaps showcases a significant shortcoming of current quantization, as most of the net change comes from diminishing capabilities of certain salient modules in Figure 3. We will add this discussion on all aspects of importance shift in our revised draft.

---

> > > ### Author Response · Authors · 2025-11-24
> > > **Response to Reviewer ST2S (Part 3)**
> > >
> > > **Weakness 3 on pruning analysis**: As mentioned in line 265, we require a pruned LRM that remains usable for interpretability analysis, as annotating and interpreting outputs that consist largely of gibberish would be meaningless. Therefore, in order to provide more insights on pruning, we first run additional benchmarking using a different pruning method called AlphaPruning suggested by Reviewer SXNL. Specifically, AlphaPruning computes layerwise compression ratios, and we apply SparseGPT to prune the layers accordingly. Because SparseGPT with 50% sparsity produces unusable outputs (especially on AIME 2024; see Table 1), we turn to AlphaPruning to assess whether the scores can be improved. We show AlphaPruning scores (averaged over 3-pass) on two R1 distilled models below.
> > >
> > > | Model                         | Compression   | AIME 2024 | FOLIO | Temporal | Avg  | MuSiQue       |
> > > |------------------------------|--------------|----------:|------:|---------:|-----:|--------------:|
> > > | DeepSeek-R1-Distill-Llama-70B | 50% sparsity |     26.7  | 74.2  |    97.7  | 66.2 | (5.3, 12.39)  |
> > > | DeepSeek-R1-Distill-Llama-8B  | 50% sparsity |      6.7  | 61.7  |    79.6  | 49.3 | (0.0, 2.95)   |
> > > | DeepSeek-R1-Distill-Llama-8B  | 30% sparsity |     41.1  | 68.9  |    82.1  | 64.1 | (0.3, 4.51)   |
> > >
> > > First, compared to SparseGPT on the R1 distilled Llama-70B, AlphaPruning achieves higher scores on most benchmarks. However, a score of 26.7 on AIME 2024 still indicates a significant level of gibberish. We then run it on R1 distilled Llama-8B under two sparsity levels. Similarly, 50% sparsity still produces unusable outputs, so we iteratively reduce the sparsity level and ultimately settle on 30%, where much stronger reasoning performance is achieved. Although 30% sparsity (1.4x reduction in size) offers much smaller compression ratio than 4-bit quantization (4x reduction in size), its “avg” score of 64.1 is significantly higher than 4-bit GPTQ and GPTAQ. Therefore, we select AlphaPruning at 30% sparsity on the R1-distilled Llama-8B as the pruned LRM for our interpretability analysis. We show the corresponding heatmaps here: https://drive.google.com/file/d/18YnqChpo5QHXxExyBVTT3D6RinRGtii3/view?usp=sharing.
> > >
> > > We observe that the heatmaps of AlphaPruning at 30% appears very similar to 4-bit AWQ on R1 distilled Llama-8B (Figure 3), 4-bit GPTQ on R1 distilled Llama-8B (Figure 7), and 4-bit AWQ on R1 distilled Qwen-7B (Figure 6). Therefore, our key finding 3 can be nicely generalized to AlphaPruning as well: AlphaPruning overly compresses the final-layer modules and MLP gate projections. This additional experiment shows the effects of pruning and the generalizability of our key finding 3 on pruning methods.

---

> > > > ### Author Response · Authors · 2025-11-24
> > > > **Response to Reviewer ST2S (Part 4)**
> > > >
> > > > **Weakness 3 on non-R1 families**: We are also excited to share that our findings generalize well to “Pinkstack/DistilGPT-OSS-qwen3-4B” (a distilled model from both GPT-OSS-20B and GPT-OSS-120B via supervised fine-tuning). In order to systematically investigate weight importance (such as the left part of Figure 2 and Figure 4) and quantization effect (such as Figure 3, 6, and 7), we interpret this distilled model and its 4-bit GPTQ quantized variant. We first show their one-pass benchmarking scores below. It is clear to see that MuSiQue scores are not affected by quantization, further implying that preserving weight count with lower precisions is a reasonable strategy for LRMs’ knowledge memorization (key finding 1).
> > > >
> > > > | Model                                         | AIME 2024 | FOLIO | Temporal | Avg  | MuSiQue     |
> > > > |----------------------------------------------|----------:|------:|---------:|-----:|------------:|
> > > > | DistilGPT-OSS-qwen3-4B                      |     36.7  | 82.3  |    94.4  | 71.1 | (4.0, 6.73) |
> > > > | 4-bit GPTQ quantized DistilGPT-OSS-qwen3-4B |     26.7  | 76.4  |    86.4  | 63.2 | (4.0, 6.92) |
> > > >
> > > > Then, we look at our key finding 2. After running our fine-grained interpretation analysis on DistilGPT-OSS-qwen3-4B, we present its heatmaps here: https://drive.google.com/file/d/1mfrAp0UHnTyujwvsJY2LR0IlQUVn2kGj/view?usp=sharing. We see that 36_up (up_projection in the final layer) is either the most important or the second most important component across four heatmaps, which demonstrates the generalizability of our key finding 2. Moreover, whenever 36_up is ranked second on a heatmap, 36_gate is always ranked first. As both up_projection and gate_projection in the final layer are important to R1 distilled models in Figure 2 (left part) and 4, we see a high similarity of weight importance between R1 distilled models and this GPT-OSS distilled model, which indicates the possibility of finding common important components across different LRMs. On the other hand, it is worth noting that different LRMs might end up with slightly different heatmaps. For example, last-layer k_projection is highlighted in Figure 4 but appears to be less important for the GPT-OSS distilled model. This shows that our interpretation can produce customized analysis of weight importance. With more discussion offered in our paper, our key finding 2 aims to locate the import weight component that is shared by different LRMs (e.g., different model sizes and classes).
> > > >
> > > > Finally, for our key finding 3, we demonstrate its generalizability through the 4-bit GPTQ quantized DistilGPT-OSS-qwen3-4B. The heatmaps for visualizing decrease of relative importance are shown here: https://drive.google.com/file/d/1QeeKS3K5IHpF1EuTVOj2bDQIkQ1xCV4R/view?usp=sharing. Most salient decreases (outliers) come from the last-layer components across all four heatmaps, along with a few salient decreases in the second last layer. Out of a small number of outliers in a heatmap, gate_proj also has at least 2 outliers on the backtracking and uncertainty estimation capabilities. Therefore, even when we switch to a GPT-OSS distilled model, GPTQ still overly compresses the final-layer modules and MLP gate projections.
> > > >
> > > > We will add our generalization experiment on AlphaPruning and non-R1 distilled LRM in our revised draft.

---

> > > > > ### Author Response · Authors · 2025-11-24
> > > > > **Response to Reviewer ST2S (Part 5)**
> > > > >
> > > > > **Question 3**: Thanks for your suggestion! For selective protection trade-offs, we present the table below that follows the same setting as Section 5.2 (3-bit AWQ with selective full-precision). Excluding table header, the performance scores of the first and the fifth row are copied from Table 4, and the last row contains the one-pass scores of R1-Distill-Llama-8B. “Protected %” shows the percentage of protected weights over all in R1-Distill-Llama-8B, and we also calculate the theoretical GPU memory consumption for loading all weights as a measure of memory overhead (with selective protection, protected weights will be loaded into 16-bit and the rest will remain in 3-bit). We show “final-layer gate vs. up vs. both” along with more protected components to reach around 4.9% protection ratio. We are not able to show inference speedup, as mix-precision inference in this case typically requires nontrivial systems work to show meaningful speedups, which is out of the scope of our paper.
> > > > >
> > > > > | Full-precision anywhere?                                  | Protected % | Memory | AIME 2024 | FOLIO | Temporal | Avg   |
> > > > > |-----------------------------------------------------------|------------:|--------------------|----------:|------:|---------:|------:|
> > > > > | -                                                         |         0   | 3.0 GB             |     10.0  | 59.6  |    68.4  | 46.00 |
> > > > > | Final-layer gate                                          |       0.7%  | 3.1 GB             |     26.7  | 64.0  |    76.0  | 55.57 |
> > > > > | Final-layer up                                            |       0.7%  | 3.1 GB             |     16.7  | 59.6  |    54.8  | 43.70 |
> > > > > | Final-layer gate & up                                     |       1.5%  | 3.2 GB             |     13.3  | 60.1  |    51.6  | 41.67 |
> > > > > | Final-layer MLP                                           |       2.2%  | 3.3 GB             |     16.7  | 67.0  |    74.0  | 52.57 |
> > > > > | Final-layer & 15_gate & 16_gate & 17_gate & 18_gate       |       4.9%  | 3.7 GB             |     20.0  | 64.5  |    69.6  | 51.37 |
> > > > > | All weights                                               |       100%  | 16 GB              |     36.7  | 70.4  |    77.6  | 61.57 |
> > > > >
> > > > > We see that memory consumption steadily increases as we protect more model components. We also notice large variability on reasoning performance, since scores do not always increase on a benchmark when we protect more weights. One reason is that score variability is greater on a smaller model. For example, according to Temporal (a relatively easier task than AIME 2024) scores in Table 1, scores of different compression strategies do not vary a lot on R1-Distill-Llama-70B, while scores of only 4-bit quantization can vary a lot more (from the minimum of 65.9 to the maximum of 88.7) on R1-Distill-Llama-8B. Another reason is that our selective protection only protects certain components, but the interpretation results show much more components that are overly compressed with various levels of severity. Therefore, since we know the top candidates for protection based on Figure 3, we are able to show a few cases (3 out of 5 cases, excluding the last row and no selective protection) when the improvement of average benchmark scores is larger than the protection percentage. This further indicates the utility of our fine-grained interpretation analysis.
> > > > >
> > > > > **Question 4**: Within the 2.51-bit LRM, the embedding matrices, output/head, and normalization/router layers are kept at higher bit widths, while almost everything else is quantized aggressively. Specifically, the embedding matrix is in 4-bit and the final language model head is in 6-bit, while all MoE (Mixture of Experts) routing matrices and layer-norm weights are left in full precision (32-bit). The majority (~88%) of weights – namely the bulk of the MoE weights – are quantized down to 2.51-bit.
> > > > >
> > > > > We do not find precise details of the calibration dataset used for 2.51-bit quantization. However, we see that Unsloth plans to use more than 1.5 million tokens for future quantized models, so we can make a reasonable prediction that the dynamic quantization uses more calibration data than other quantization methods we benchmark in our paper. We adopt the same calibration data for all other quantization. The dynamically quantized models work on MoE architecture such as R1, while other quantization methods (e.g., AWQ and GPTAQ) are designed for non-MoE with less weight count. Therefore, due to different compression targets, we are not able to compare dynamic quantization and other methods under identical calibration. In Table 1, we aim to report various methods as comprehensive as possible in order to study the effects of compression on reasoning performance.

---

> ### Comment · Reviewer_ST2S · 2025-11-27
>
> Thanks for the detailed response, which addresses my questions. I would keep my score.

---

> > ### Author Response · Authors · 2025-11-27
> >
> > Thanks so much for your acknowledgment! We are glad that your questions have been addressed. If you feel it is appropriate, we would appreciate it if you could consider an increase to your rating to support this paper. 😊 We are also more than happy to address any additional concerns you may have before the deadline.

---

### Author Response · Authors · 2025-12-03
**Our draft has been revised to incorporate all comments.**

We sincerely thank all the reviewers for their valuable feedback. We have incorporated their comments in our revised draft. Most modifications are in the appendix with appropriate pointers (e.g., mentions) in the main content. We just updated our PDF. Below are the major modifications:

1. We added our human validation results in Appendix G to demonstrate annotation robustness.

2. In Appendix H, we elaborated our justification of only visualizing the decrease of relative importance in the main content.

3. We added pruning effect in Appendix I to showcase the generalizability of our third key finding on pruning methods.

4. We added the performance scores of AlphaPruning in Table 1.

5. We demonstrated the generalizability of our findings on non-R1 families in Appendix J.

6. We described the 2.51-bit R1 with key details in Appendix D.

7. We added the discussion and the visualization on importance scores of the base model (Llama-3.1-8B) in Appendix K.

8. We added our experiments on naive RAG to decouple reasoning from knowledge memorization in Appendix L.

9. We added AWQ as another diagnosis to show the alignment between activation signals and our key finding 2 in Appendix M.

10. We added our discussion on module-wise compression and identifying weights important to reasoning in Appendix B.5.

11. We made Figure 2 larger by utilizing more vertical space. **Thus, the “left” and “right” parts of Figure 2 used during the rebuttal now correspond to the upper and lower parts in the revised draft.**

12. We added additional selective quantization experiments on AIME 2024 to demonstrate the lower output quality of only quantizing 32_up  in Appendix N.

13. We added additional analysis of test-time compute and model collapse behavior in Appendices O and P to show our good-faith effort of understanding model behavior.

---

### Author Response · Authors · 2025-12-03
**Final Summary for Area Chair Review**

We sincerely thank the AC for considering our rebuttal discussion! Even though the discussion phase was closed early, we still received acknowledgements (without any follow-up questions) from 3 out of 5 reviewers who claimed that their concerns or questions had been addressed. Notably, Reviewers SXNL and ja42 mentioned that they would “raise” or “adjust” their ratings based on our detailed responses, so it is reasonable to assume that their overall ratings would be at least 6. Reviewer ST2S maintained the original score of 6. We also note that the remaining two reviewers (PAMX and G6hk) share similar concerns, so we believe their concerns are also fully addressed. We believe the scores will be 6/6/6/6/6 if a full discussion period had occurred.

Overall, we believe all reviewers’ concerns are addressed. These are common concerns shared by at least 2 reviewers: annotation robustness (ST2S, PAMX, SXNL), generalizability of our findings on non-R1 families (ST2S, PAMX, SXNL, ja42, G6hk), lack of interpretation on pruned LRMs (ST2S, PAMX, ja42), importance scores of the base model Llama-3.1-8B (PAMX, SXNL), a single typo (ja42, G6hk), and decoupling reasoning from knowledge memorization (PAMX, ja42). During the discussion phase, we **demonstrated** annotation robustness with additional human validation, **extended** all of our findings to non-R1 by running experiments on DistilGPT-OSS-qwen3-4B, **enriched** our pruning analysis by providing additional benchmarking and visualization of AlphaPruning, **reinforced** Takeaway 4.3 by visualizing importance scores of the base model (Llama-3.1-8B), **fixed** the single typo, and **decoupled** reasoning from knowledge memorization by running an additional RAG pipeline. **These common concerns are confirmed as resolved by at least one reviewer.**

We believe other specific comments were also fully addressed by our point-by-point responses. Specifically, here are the details of how we addressed the unique concerns of each reviewer:

- ST2S: We **justified** our visualization of score decrease by showing heatmaps of other cases with more explanation, **expanded** our selective protection with additional trade-off analysis, and **described** the 2.51-bit R1 with key details.

- PAMX: All concerns are common ones elaborated above.

- SXNL: We **addressed** the concern on “strong claim” by extending all of our findings to non-R1 and drawing connections to equations in Sections 2.2 and 2.3, **diversified** our diagnosis by leveraging intermediate AWQ results, and **incorporated** 3 relevant papers by offering our insights on weight-only compression.

- ja42: All concerns are common ones elaborated above.

- G6hk: We **showcased** the lower output quality of only quantizing 32_up on AIME 2024 by inspecting gibberish outputs and performing additional selective quantization, **resolved** the concern on Figure 2 by making the figure larger, and **confirmed** the invariance of our identified important components over the change of task type/difficulty by clarifying key details in our paper.

Finally, as promised, we have incorporated all comments in our revised draft. Modifications are specified in another post.

---

### Meta-Review · Area_Chair_hm7N · 2026-01-02

**Summary:**

**Paper Summary:**
The paper systematically analyzes how compression methods—quantization, distillation, and pruning—affect the reasoning capabilities of large reasoning models (LRMs) through benchmarking and mechanistic interpretability, revealing critical components and proposing strategies to mitigate performance loss.

**Strengths:**

1. Comprehensive scope: It evaluates three major compression families across multiple reasoning datasets (AIME 2024, FOLIO, Temporal Sequences, MuSiQue) with clear comparative tables.
2. Novel interpretability application: It uses fine-grained mechanistic analysis (difference-of-means and attribution patching) to identify reasoning-critical components.
3. Actionable insights: It demonstrates that protecting ~2% of weights (final-layer MLPs) during quantization recovers significant accuracy (+6.57%).
4. Strong empirical validation: It includes selective quantization experiments and generalization to non-R1 models, supporting robustness of findings.
5. Clear presentation: Well-structured narrative and readable figures, with practical recommendations for compression design.

**Weaknesses:**

1. Limited generalizability: The analysis focuses mainly on DeepSeek-R1 and its distilled variants; broader validation on diverse LRM families is limited.
2. Questionable annotation robustness: The behavior labeling relies on GPT-4o prompts, introducing potential subjectivity and bias.
3. Imbalanced coverage: Quantization is deeply analyzed, while pruning receives minimal mechanistic interpretation beyond collapse points.
4. Lack of statistical rigor: The paper lacks uncertainty measures (variance, confidence intervals) and rank-stability checks for importance scores.
5. Interpretability gaps: The paper does not fully explain why certain deep layers are critical for reasoning; relies on a single interpretability framework without methodological diversity.

**Reviewer Concerns:**

Most of the questions have been responded and discussed by the authors. The revision has addressed most concerns.

**Reviewer Scores:**

I think a few reviewers could increase the rating from 4 to 6.

---

### Decision · Program_Chairs · 2026-01-26

Accept (Poster)